

# The Paris Low-Level Jet During PANAME 2022 and its Impact on the Summertime Urban Heat Island

Jonnathan Céspedes[1,2], Simone Kotthaus[1], Jana Preissler[2,3], Clément Toupoint[2], Ludovic Thobois[2], Marc-Antoine Drouin[4], Jean-Charles Dupont[5], Aurélien Faucheux[6], and Martial Haeffelin[7]

[1]Laboratoire de Météorologie Dynamique (LMD-IPSL), CNRS, École Polytechnique, Institut Polytechnique de Paris, 91128 Palaiseau Cedex, France
[2]Vaisala France SAS, 6A rue René Razel, 91400 Saclay, France
[3]Dhara Consulting Services, Darmstadt, Germany
[4]LMD/IPSL, École Polytechnique, Institut Polytechnique de Paris, ENS, PSL Research University, Sorbonne Université, CNRS, Palaiseau France
[5]Institut Pierre Simon Laplace (IPSL), Université Versailles Saint-Quentin-en Yvelines, Palaiseau Cedex, France
[6]CEREA, École des Ponts ParisTech, EDF R&D, IPSL, Marne-la-Vallée, 77455, France
[7]Institut Pierre Simon Laplace (IPSL), CNRS, Palaiseau Cedex, France

**Correspondence:** Jonnathan Céspedes (jonnathan.cespedes@lmd.ipsl.fr)

**Abstract.**

The Low-Level Jet (LLJ) and the Urban Heat Island (UHI) are common nocturnal phenomena in the atmospheric boundary layer. While the canopy layer UHI has been studied extensively, interactions of the LLJ and the urban atmosphere in general (and the UHI in particular) have received less attention. In the framework of the PANAME initiative in the Paris region, continuous measurements of wind speed and vertical velocity profiles were recorded with two Doppler Wind Lidars (DWL) - for the first time allowing for a detailed investigation of the summertime LLJ characteristics in the region. Jets are detected for 70% of the examined nights, often simultaneously at an urban and a suburban site highlighting the LLJ regional spatial extent. Emerging at around sunset, the mean LLJ duration is $\sim$10 h, the mean wind speed is $\sim$9 m s$^{-1}$, and the average core height is 400 m above the city. For many jets, results show signatures in the temporal evolution that indicate the inertial oscillation mechanism plays a role in the jet development: a clockwise veering of the wind direction and a rapid acceleration followed by a slower deceleration. The LLJ core induces variance in the vertical velocity ($\sigma_w^2$) above the urban canopy layer. It is shown that $\sigma_w^2$ is a powerful indicator for the air temperature spatial contrasts as UHI intensity decreases exponentially with increasing $\sigma_w^2$ and strong values only occur when $\sigma_w^2$ is very weak. This study demonstrates how DWL observations in cities provide valuable insights into near-surface processes relevant to human and environmental health.





# 1 Introduction

The nocturnal Urban Heat Island (UHI) in the canopy layer is surely among the most studied phenomena in the urban environment. As this UHI is defined as the difference in air temperature between built-up and rural settings right above ground level (Oke et al., 2017), investigations usually focus on the near-surface atmosphere conditions, while the dynamics of the
Atmospheric Boundary Layer (ABL) are rarely considered explicitly. Although wind speed and atmospheric stratification play a significant role in the formation of the UHI, the intensity of the UHI is strongest under low wind speed and cloud-free conditions (Oke et al., 2017), i.e., when atmospheric stratification in rural settings tends to be relatively stable. During stable atmospheric stratification, the atmospheric boundary layer flow may become decoupled from the friction exerted by the surface, which can lead to the formation of the nocturnal Low-Level Jet (LLJ). The LLJ is manifested by a sharp maximum in the vertical
profile of the horizontal wind speed, typically at a height between $100\,\mathrm{m}$ and $1000\,\mathrm{m}$ above the ground, referred to as the core of the jet (Stull, 1988). Another feature related to the LLJ wind profile is a strong decrease that goes along with a minimum in the horizontal wind speed above the core height (Shapiro et al., 2016). The LLJ is a mesoscale phenomenon frequently observed in the ABL, mostly during nights with fair-weather conditions after clear-sky days. It is usually characterized based on its core height, wind speed and wind direction. With clear links to various processes such as advection, wind shear and
turbulent mixing, the LLJ core characteristics can have a series of practical implications with respect to, e.g., air quality (Wei et al., 2023; Klein et al., 2019), changes in precipitation patterns (Algarra et al., 2019; Chen et al., 2022), aviation safety (Gultepe et al., 2019; Liu et al., 2014), the potential of, and risk to, wind energy production (Lundquist, 2021; Luiz and Fiedler, 2022; Rubio et al., 2022), urban ventilation (He et al., 2022), as well as the heat spatial distribution and intensity of the UHI effect (Kallistratova and Kouznetsov, 2012; Hu et al., 2013; Ulpiani, 2021; Lin et al., 2022). The LLJ is highly relevant for
meteorology because its core characteristics are associated with the turbulent mass exchange in the ABL (Blackadar, 1957).

Since the 1950s, the LLJ has been extensively documented worldwide. In The Great Plains, USA, it is observed throughout the year (Blackadar, 1957; Helfand and Schubert, 1995; Higgins et al., 1997; Banta et al., 2002; Shapiro et al., 2016). Also, it has been frequently observed in China (Zhang et al., 2019; Du and Chen, 2019); Finland (Tuononen et al., 2015, 2017); The Netherlands (Baas et al., 2009; van de Wiel et al., 2010); and more recently in South America (Jiménez-Sánchez et al., 2019;
Jones, 2019; Builes-Jaramillo et al., 2022; Sánchez et al., 2022), Africa (Chakraborty et al., 2009; Hartman, 2018), Southeast Asia (Li and Du, 2021; Chen et al., 2022), and Australia (Qi et al., 1999). One converging point of the previous studies is that the LLJ core generally is formed during the evening, followed by an increase in the core wind speed, and it eventually dissipates at about sunrise. Blackadar (1957) proposed the inertial oscillation theory to explain the physical mechanism behind the LLJ formation, but other mechanisms can also play a role. In The Great Plains, the LLJ formation is linked to the IO and the
differential heating and cooling of the sloping terrain (Holton, 1967; Shapiro et al., 2016). Jets are also observed over coastal regions as the result of the land–sea breeze interactions and temperature gradients (Karipot et al., 2009; Roy et al., 2021). However, regardless of the study area or mechanism of formation, most of the observational-descriptive studies agreed on the importance of classifying the LLJ, usually by one of its main characteristics (Banta et al., 2002; Baas et al., 2009; Karipot et al., 2009; Bonin et al., 2015). Grouping of LLJ events into categories facilitates description and the analysis of potential effects on





the environment of the affected area. The most widely used classification criterion is the core wind speed. Banta et al. (2002) were one of the first to propose four categories of wind speed to describe LLJs: (0-5), (5-10), (10-15) and (15-20) $\mathrm{m\,s^{-1}}$. Subsequently, several studies around the world have adopted similar categories of wind speed to study the LLJ (Karipot et al., 2009; Kallistratova et al., 2013; Wei et al., 2013; Arfeuille et al., 2015; Vanderwende et al., 2015; Carroll et al., 2019). This classification system provides a better understanding of the LLJ phenomenon development, evolution, and impacts based on the

wind speed intensity. Studies focused on the effects of the LLJ dynamics on the near-surface atmosphere, tend to consider other parameters like Turbulent Kinetic Energy (TKE), in addition to the core wind speed. Banta et al. (2002, 2003) successfully showed that the mechanical turbulence generated by the wind shear below the jet core plays a key role in controlling fluxes between the surface and the atmosphere. In particular, the downward transport of turbulence can modulate near-urban surface atmospheric processes like the UHI effect (Hu et al., 2013; Lin et al., 2022).

The UHI can be observed worldwide throughout the year, but in high-density cities in the mid-latitudes, it tends to be stronger during the summer (Lemonsu and Masson, 2002; Oke et al., 2017). The expansion of urban areas under poor ventilation and cooling design has made the UHI a prominent factor in the deterioration of the human thermal comfort (Kong et al., 2016; Li et al., 2019; Lin et al., 2022). High nocturnal temperatures increase the stress on the human body at times when it needs to rest (sleep), leading to higher mortality (Robine et al., 2008; Taylor et al., 2015; Ridder et al., 2016; He et al., 2022). While

the UHI phenomenon is driven by access heat in the built environment, atmospheric dynamics (advection, mixing, subsidence) significantly modulate the UHI intensity (Oke, 1973; Oke et al., 2017). Weak flow conditions favor the formation of strong UHI (Lemonsu and Masson, 2002), while each increase of $1\,\mathrm{m\,s^{-1}}$ in wind speed was found to reduce the urban air temperature during summer nights by up to $2\,^{\circ}\mathrm{C}$ (Cheng et al., 2012; He et al., 2022).

   Although surface wind speed has long been recognized as an important indicator for variations in the UHI intensity, only a
few studies in urban areas have investigated the relation between advection produced by the nocturnal LLJ and the UHI. Studies conducted in Sao Paulo, Brazil (Sánchez et al., 2022); Oklahoma, USA (Hu et al., 2013); Moscow, Russia (Kallistratova and Kouznetsov, 2012; Kallistratova et al., 2013); and Beijing, China (Lin et al., 2022); concluded that there is a negative correlation between the UHI intensity and the core wind speed of the jet. Even though these efforts have contributed to understanding the potential implications of the LLJ on UHI development, these works are mostly based on a limited number of radiosonde

observations and few case studies. Hence, continuous observations of the wind speed profile with high temporal and spatial resolution are still needed in urban environments. Another relevant aspect of the interaction between the LLJ and the Urban Boundary Layer (UBL) is that the urban environment enhances atmospheric mixing and buoyancy that can again also affect the characteristics of those LLJs originated in the rural surroundings. In London, UK, Barlow et al. (2014) and Tsiringakis et al. (2022), have investigated the interaction between the LLJ and the UBL based on a combination of continuous DWL

observations and modeling data, highlighting the importance of both the LLJ (downward mixing by shear-driven turbulence) and the urban surface (upward mixing driven by urban heat and roughness).

   In France, the LLJ has been observed in northern coastal urban areas (Roy et al., 2021, 2022; Dieudonné et al., 2023), but coastal jets usually present different formation mechanisms and characteristics than those observed inland. This is mainly explained due to the importance of the sea-land breeze interactions (Karipot et al., 2009). Particularly in the urban area of





Paris, some studies have been conducted to understand the impact of the LLJ on air quality, the mixing of pollutants within the UBL, and the UHI development. Klein et al. (2019) performed a one-day case study using a combination of DWL profiles, numerical simulations and ancillary observations, concluding that nocturnal LLJs can modulate the evolution of the mixing layer with implications for the ozone concentration in the early morning. Cheliotis et al. (2021) collected DWL data during a period of 3 months in fall 2014, observing LLJs in 20 out of 63 nights. They linked the production of turbulent coherent

structures to the presence of LLJ events, that in turn play an important role in the transport of heat, moisture and pollutants through the ABL. Wouters et al. (2013) presented a model-based case study during the Summer of 2006, where simulations of UHI and LLJ allowed them to conclude that the UHI could be affected by the nocturnal stability and mixing associated to the LLJ. However, when comparing with radiosonde profiles the strong jets were underestimated by the model, making it hard to identify what characteristics of the jets influenced the UHI development. Therefore, as far as the authors of the present

work are aware, variations of the LLJ characteristics over periods exceeding a few days and their possible implications for UHI development have not yet been investigated in the Paris region. One key reason is the previous lack of medium or continuous long-term observations of the wind speed profile within the Paris UBL.

Strengthening the knowledge in this field is important because there is observational evidence that the LLJ is a common phenomenon in Europe, frequently detected at distances of less than $400\,\mathrm{km}$ from the city of Paris (Baas et al., 2009).

Furthermore, gaining understanding about the LLJ impacts on the UHI is imperative, given the significance of UHI as a main concern in a city like Paris where a large number of inhabitants (approx. 12 million) is potentially exposed to severe heat hazards that are expected to become increasingly severe in future climates. As a consequence, the present study is going to investigate the characteristics and nocturnal temporal evolution of the LLJ observed over the Paris region in summer 2022, and what are their implications for the UHI intensity. For the first time, providing a comprehensive description of the occurrence of

summertime LLJ in the Paris region and an analysis of its variability. Here, we assess the impact of the LLJ on the profile of the horizontal wind speed and the UHI intensity. This paper is structured as follows: Section 2 provides details of the study area, study period, DWL and surface-based meteorological observations, and automatic detection of LLJ and UHI determination; Section 3 presents the results of the LLJ detection and performance, LLJ classification, LLJ characteristics and wind profile, LLJ nocturnal evolution, and the UHI evolution; finally, Section 4 present the summary and conclusions of this paper.

## 2 Methods and materials

### 2.1 Study area and study period

Here we study the LLJ characteristics in the Paris region, France. With a population of over 12 million people, the Paris agglomeration is the second biggest megalopolis and the most densely populated city in Europe. The Paris region has experienced fast and wide urbanization during the last decades, but is still surrounded by natural and agricultural areas. Anthropogenic heat

emissions and changes in land cover are aggravating the UHI effect (Lemonsu and Masson, 2002). Fig. 1 shows the topography of the Paris region, the land cover classification, and the geographical location of the experimental sites. Paris is located about $150\,\mathrm{km}$ from the sea, in a valley defined by the Seine River basin and surrounded by plateaus with modest topographic variation,




which rise no more than 230 m above sea level (asl). In the city of Paris, the terrain is relatively flat, the lowest altitude is about
20 m asl along the river, and the highest is about 130 m asl on hills in the northern parts of the city. About 20 km southwest
of the city, the Paris-Saclay Plateau is located, which has a mean extension of 10 km and is elevated about 160 m asl. It is an
important economic, academic, and industrial area in the region. Due to the distance from the sea, there is no interaction of the
sea breeze with the urban area. The long-term prevailing wind direction in the area is southwesterly and has a maritime origin,
but winds from the northeast are also often observed (Haeffelin et al., 2005; Pal et al., 2012).

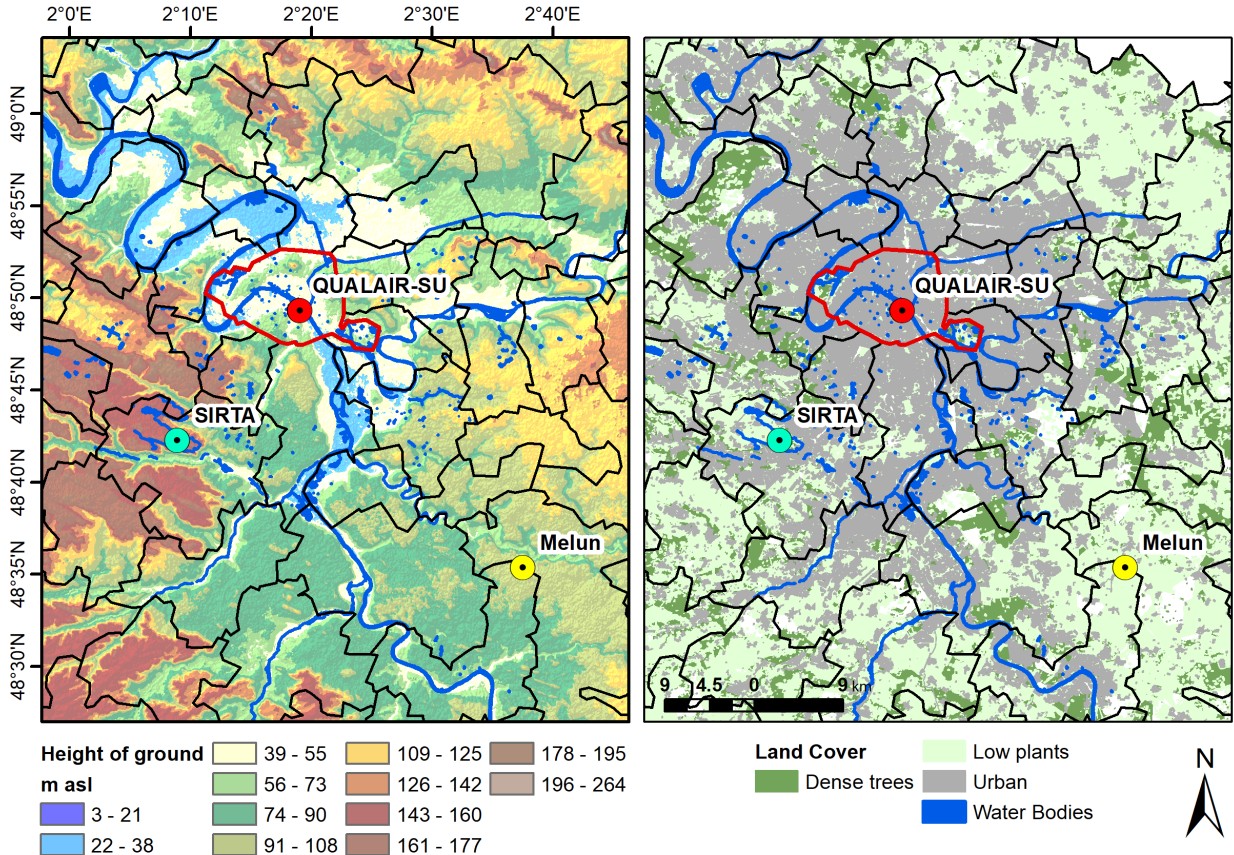

**Figure 1.** Paris region (France) study area and the location of the experimental sites. a) Topographic map from ESA (2023), and b) land cover
map from Region (2023). The red borders indicate the administrative boundary of the City of Paris. The black polygons are administrative
borders within the Paris region. The red dot represents the location of the WindCube Scan 400S at QUALAIR-SU (urban lidar), the cyan
dot is the location of the WindCube WLS70 at SIRTA (suburban site), and the yellow dot is the location of Météo France meteorological
measurement site at Melun (rural site).

Using a comprehensive data set spanning from 2006 to 2022, collected by the Météo France surface meteorological network
and preprocessed by the ReOBS project (Chiriaco et al., 2018), the climatology of synoptic conditions within the study area





is analyzed for the summer months of June, July, and August. The prevailing nocturnal winds exhibit a range of direction mostly between $170°$ and $45°$, with speeds below $8\,\mathrm{m\,s^{-1}}$. The nocturnal air temperature fluctuates within the range of $6.6\,°C$ to $38\,°C$, with a mean temperature of $18.6\,°C$. This study focuses on the summer of 2022, between 15 June and 31 August. The year 2022 was recorded as the warmest observed in France so far, with three reported heat waves between June and August,

leading to an exceptional nocturnal mesoscale flow over the Paris region during summertime with a predominant northeasterly flow. Severe conditions were observed especially in the July heat wave, where sea-level temperature anomalies above $12\,°C$ were associated with a deficit in precipitation rate of about 80%, leading to extreme droughts (Petit et al., 2023). In general, the summer of 2022 had prevailing winds coming from the northerly to northeasterly directions ($0°$-$75°$), with speeds below $7\,\mathrm{m\,s^{-1}}$. The air temperature ranged between $11.8\,°C$ and $35.2\,°C$, with a mean value of $20.6\,°C$.

## 2.2 Doppler Wind Lidar measurements

The high-power scanning Doppler Wind Lidar (DWL) Vaisala WindCube Scan 400S was deployed in the city center of Paris (red dot in Fig. 1), on the roof of the Zamansky tower ($88\,\mathrm{m}$ above ground level (agl)) at the QUALAIR-SU supersite ($48.8°N$; $2.36°E$; $125\,\mathrm{m}$ asl). QUALAIR-SU is a platform dedicated to atmospheric observations and air quality monitoring, which is operated by the Laboratoire Atmosphères, Observations Spatiales (LATMOS) and is hosted by Sorbonne Université, at the

Jussieu campus (Qualair, 2022). The WindCube Scan 400S is a DWL equipped with a scanning head capable of orienting the laser beam in any direction of the hemisphere (Thobois et al., 2019; Dolfi-Bouteyre et al., 2008). The high energy emitted by the pulse, at $1.54\,\mathrm{\mu m}$ wavelength, allows it to sample the atmosphere at a distance of up to $7\,\mathrm{km}$. Several configurations of pulses are available corresponding to different spatial resolutions (75, 100, 150, or $200\,\mathrm{m}$) with a resolution of $75\,\mathrm{m}$ used in this study. A blind zone, spanning twice the spatial resolution ($150\,\mathrm{m}$ in this study), restricts measurements close to the

sensor. Table 1 provides a summary of the main properties of the instrument. The WindCube Scan 400S can measure under different scanning strategies such as vertical stare, Plan-Position Indicator (PPI) and Doppler Beam Swinging (DBS). Liu et al. (2019) provide a detailed technical description of every scan mode operation and their potential applications. The DBS mode is widely used in urban meteorology to measure vertical profiles of horizontal wind speed and wind direction. It was shown by Pearson et al. (2009) that DBS operates fast and it well-capture unstable flows in the urban interface. In this study, a *five-point*

DBS mode is used, with the scanning head successively addressing five lines of sight (LOS): one vertically oriented LOS ($90°$ elevation angle) and four tilted LOS ($75°$ elevation angle) pointing north, east, south, and west, respectively. Each scanning sequence (1 DBS cycle) is completed in approximately $15\,\mathrm{s}$, with $1\,\mathrm{s}$ of accumulation time for each LOS and $2\,\mathrm{s}$ between two LOS. The WindCube Scan 400S has been aligned with the geographic north, using the hard target method which has a precision of $\pm2°$.

During the study period (15 June - 31 August 2022) the scan strategy was designed to follow an hourly schedule, integrating the DBS scan with the vertical stare and PPI. During the first and third quarter of every hour, the DBS was performed for $9\,\mathrm{min}$ followed by a full circumference PPI at $0°$ of elevation with an angular resolution of $2°/s$, and finally $3\,\mathrm{min}$ of continuous vertical stare at $90°$ of elevation. The second and fourth quarter begins with $2\,\mathrm{min}$ of vertical stare at $90°$ of elevation, followed by $9\,\mathrm{min}$ of DBS, and ends with a full PPI at $0°$ of elevation. In this study, all vertical profiles of horizontal wind speed





**Table 1.** Properties of the Vaisala Doppler Wind Lidars WindCube Scan 400S (Urban site) and WLS70 (Suburban site). For the WindCube Scan 400s, the symbol **(*)** marks the setting used in this study.

| Properties | WindCube Scan 400S (Urban) | WindCube WLS70 (Suburban) |
| --- | --- | --- |
| Altitude Location (m asl) | 125 | 155 |
| Pulse repetition frequency (kHz) | 7, 10*, 20, or 40 | 10 |
| Pulse width (ns) | 100*, 200, 400, or 800 | 100 |
| Range gate resolution (m) | 75*, 100, 150 or 200 | 50 |
| Min range (m) | 150 | 100 |
| Max range (m) | 7000 | 4000 |
| Accumulation time (s) | 1 per beam | 1 per sequence |
| Radial wind speed range ($\mathrm{m\,s^{-1}}$) | $\pm 30$ | $\pm 30$ |
| Emission wavelength (µm) | 1.54 | 1.54 |
| Radial wind accuracy ($\mathrm{m\,s^{-1}}$) | 0.1 | 0.3 |
| Height location (m agl) | 88 | 5 |
| First available gate (m agl) | 238 | 105 |

observations were derived from the DBS, while the vertical stare data are used to derive the Vertical Velocity Variance ($\sigma_w^2$). Observations from the PPI are not used in this study. The implemented scanning schedule was designed to fulfill several objectives but was not specifically optimized for the detection of LLJ. One quality control step is implemented to ensure a high-quality wind profile product, data with CNR (Carrier-Noise-Ratio) below $-20\,\mathrm{dB}$ and above $5\,\mathrm{dB}$ are excluded whereby omitting weak signals and clouds, respectively.

In addition to the observations in central Paris, data collected by a Vaisala WindCube WLS70 at a suburban location on the Plateau Saclay are used in this study. This profiling DWL is an instrument specially developed for meteorological applications (Cariou et al., 2009). The observations are conducted at the SIRTA observatory (Site Instrumental de Recherche par Télédetection Atmosphérique (Haeffelin et al., 2005)), located on the campus of Ecole Polytechnique in Palaiseau, $20\,\mathrm{km}$ southwest of Paris ($48.713°$N; $2.208°$E; $156\,\mathrm{m}$ asl; Fig. 1). Over the last 25 years, SIRTA has collected a comprehensive

data set of atmospheric observations using in-situ measurements as well as passive and active remote sensing instruments, characterizing the regional atmospheric background of the Paris region (Dupont et al., 2016). The WindCube WLS70 uses a $1.54\,\mu\mathrm{m}$ pulsed fiber laser and a coherent detection system which provides sufficient backscattering signal up to $4\,\mathrm{km}$. Every LOS performed by the WLS70 has a fixed spatial resolution of $50\,\mathrm{m}$ (see Table 1). The WLS70 performs four-point DBS measurements ($10\,\mathrm{s}$), in a similar way to the WindCube Scan 400S, but without the vertical LOS. This study uses 10-min

averaged profiles, a previous quality control step is applied to these profiles to ensure a 80% of data availability at each range gate.



## 2.3 Automatic LLJ detection

Various methods and criteria have been developed to identify a LLJ in a vertical profile of horizontal wind, with techniques depending on the characteristics of the data source (e.g. limitations from measurement sensors) and even the study area. Stull (1988) defined a LLJ as any lower-tropospheric wind maximum in the vertical profile of horizontal wind speed, that is at least $2\,\mathrm{m\,s^{-1}}$ greater than speeds both above and below, within the lowest $1500\,\mathrm{m}$ of the atmosphere. This absolute criterion is widely found in the literature. It was applied to SODAR data collections (Karipot et al., 2009; Duarte et al., 2015) and radiosonde profiles measured in the USA (Bonner, 1968; Whiteman et al., 1997), and even in the Arctic Sea (Andreas et al., 2000). The threshold of this absolute criterion can be tuned as done by Banta et al. (2002), who used a value of $0.5\,\mathrm{m\,s^{-1}}$. However, under very weak flow conditions ($< 2\,\mathrm{m\,s^{-1}}$) the performance of the criterion could present limitations in the identification of LLJ events (Baas et al., 2009). In such cases, it is possible to use a relative threshold instead, in which a LLJ is detected by a velocity difference of 20%-25% between the local maxima and minima. In fact, absolute and relative thresholds have been combined to further improve detection. In different long-term radiosonde campaigns in Sao Paulo, Brazil, the criterion to identify a LLJ event in a single wind speed profile was that the maximum at the LLJ must be both greater than or equal to $2\,\mathrm{m\,s^{-1}}$ and 25% faster than the minimum above (Sánchez et al., 2022, 2020). Although in other DWL or SODAR-based studies, the same thresholds have been used, the high temporal and spatial resolution of surface-based remote sensing observations allows the incorporation of complementary steps. This enables not only an optimized detection of individual LLJ profiles but also the identification of LLJ events evolving over the time (Baas et al., 2009; Tuononen et al., 2015, 2017). In this study, the automatic LLJ detection is an adaptation from the method described by Tuononen et al. (2017) and uses 30-min averaged profiles of horizontal wind speed, to best combine the 15-min data products available at QUALAIR-SU and the 10-min data products available at SIRTA. This temporal resolution was previously used in SODAR studies presenting coherent LLJ detection (Karipot et al., 2009; Baas et al., 2009; Duarte et al., 2015).

Therefore, based on the literature and exploratory tests on the data set, some important conceptual aspects were adapted compared to the original method from Tuononen et al. (2017):

- Daytime LLJ events are negligible in the Paris region. During the day, very few LLJ events are recorded at the urban site. Therefore, this study focuses on the nocturnal periods, times between 18 h and 9 h UTC (20 h and 11 h local time).

- The maximum height explored in every 30-min wind profile is $1000\,\mathrm{m}$ agl.

- In the study area, 99% of nocturnal periods present only one LLJ event within the ABL. Therefore, this study considers the presence of only one single LLJ event per night, unlike Tuononen et al. (2017) who accounted for the possible development of up to three simultaneous LLJ events within the ABL column in the same night.

Additionally, technical aspects of the algorithm were also adapted. In the following, a brief description of the implemented adaptations in this study is given, further details can be found in Tuononen et al. (2017):

- **LLJ detection from a single wind profile**: we consider every horizontal wind speed profile between the first available gate ($238\,\mathrm{m}$ agl at QUALAIR-SU and $105\,\mathrm{m}$ agl at SIRTA) and $1000\,\mathrm{m}$ agl. A LLJ is identified in a profile if the local





maximum of the horizontal wind speed is at least $1.5\,\mathrm{m\,s^{-1}}$ stronger than the first local minimum above or below. Note that the minimum below the core height may not be captured correctly by the observations because no information is available $< 238\,\mathrm{m}$ agl in the instrument's blind zone. A relative detection criterion was tested, but it produced a high number of false negatives (43%).

    – **LLJ event detection**: a coherent LLJ event is defined as an event in which at least four consecutive profiles are marked
as LLJ (i.e. over course of $2\,\mathrm{h}$). In addition, the following conditions must be met between two LLJ candidate profiles:

        – difference in core height $< 150\,\mathrm{m}$,

        – difference in core wind speed $< 20\%$,

        – difference in core wind direction $< 45°$,

        – difference in core time $< 1.5\,\mathrm{h}$.

The output parameters of the algorithm are listed as follows. i) LLJ occurrence: is a Boolean value (True [1] or False [0]) given to each profile; ii) LLJ core height: height of the maximum in the wind speed profile, in this case, the algorithm can provide it both in m agl and asl; iii) LLJ core speed: value of the wind speed at the core height; iv) LLJ core direction: wind direction at the height of the wind sped maximum. Additionally, the algorithm identifies the corresponding parameters for both the minimum above and below the jet core, in case they are recorded by the profile observations.

**2.4   Surface data and UHI determination**

The UHI intensity ($\Delta$UHI $= T_{urban} - T_{rural}$) is defined as the difference in air temperature between an urban ($\mathrm{T}_{urban}$) and a rural ($\mathrm{T}_{rural}$) site. Usually, standard meteorological observations at $2\,\mathrm{m}$ agl height are compared. In the current work, the QUALAIR-SU site (red dot in Fig. 1) is chosen as the urban site, and a Météo France station at Melun (48.613°N; 2.679°E; $91\,\mathrm{m}$ asl) is chosen as the rural site (yellow dot in Fig. 1).

Data collected at the QUALAIR-SU supersite were used to represent the central urban air temperature ($\mathrm{T}_{urban}$) because continuous and long-term observations are available. During the Intensive Observation Period (IOP) in the framework of PANAME-2022 campaign, surface-based meteorological stations were installed to capture the air temperature within the built environment at street level, e.g. at Place de la Madeleine ($48.870\,84$°N; $2.324\,32$°N; $79\,\mathrm{m}$ asl) and Boulevard des Capucines ($48.870\,66$°N; $2.331\,75$°N; $81\,\mathrm{m}$ asl) on 10 July 2022, i.e. later than the start of the analysis period of the current study (15
June 2022). Hence, data from the QUALAIR-SU meteorological station located at roof level ($20\,\mathrm{m}$ agl) are here considered to well represent the intensity of the $\Delta$UHI for the Paris region. $\Delta$UHI is stronger when using the street-level urban measurement sites by about $1\,°\mathrm{C}$ when $\Delta$UHI $> 6\,°\mathrm{C}$ (see Fig. A1).

     The temperature at Melun is measured using a Sterela Opale UME at $2\,\mathrm{m}$ agl, and that at QUALAIR-SU using a Vaisala WXT520 automatic weather station at $22\,\mathrm{m}$ agl. The mean building height around the QUALAIR-SU site is about $25\,\mathrm{m}$.
30-min averages are calculated for both sites to match the DWL analysis intervals. The average nocturnal UHI intensity is calculated as the mean of temperature differences recorded during the nocturnal period (sunset to sunrise). Additionally, based



on cloud base height observations from a Lufft CHM15k automatic lidar ceilometer operated at the QUALIR-SU site, nights were classified into cloudy, partly cloudy, and cloud-free periods using the classification approach presented by Kotthaus and Grimmond (2018).

## 3 Results

### 3.1 LLJ detection performance

The time difference between two consecutive time steps for which a LLJ is detected gives an indication of the persistence of the jet event. The threshold sensitivity and performance analysis by Tuononen et al. (2017) revealed that the time-delta difference threshold has a strong influence on the performance of the algorithm. This may bias the algorithm towards the detection of false positives or very short events, which are not object of study in this work. The detection of false positives is further dependent on the quality of the measurements, which in turn depends on factors such as power and sensitivity of the laser system, vertical distribution of the aerosol load (tracers, see Section 2.2), and the presence of clouds.

An example of a coherent LLJ detection is presented in Fig. 2a. The jet core becomes visible in the late evening of the 9 August at 19h UTC at $550\,\mathrm{m}$ agl, and with a wind speed of approximately $8\,\mathrm{m\,s^{-1}}$. After sunset, the core height stabilizes at $500\,\mathrm{m}$ agl and reaches a maximum wind speed between 21h and 23 UTC. After midnight, the core height increases beyond $550\,\mathrm{m}$ agl, followed by a gradual decrease in the core wind speed. The jet persists throughout the night and into the early morning before it dissipates at 8h UTC on the 10 August. Fig. 2b presents a LLJ detection under weaker flow conditions and lower boundary layer height than the previous example. This LLJ event clearly failed the criterion of four consecutive detection explained in Section 2.2, and it is not considered in this study.

### 3.2 LLJ classification

By classifying LLJ according to their core wind speed magnitude, Banta et al. (2002) found that strong jets tend to occur at greater altitudes. Banta et al. (2003) studied the relationship between LLJ characteristics and turbulence within the stable nocturnal boundary layer, concluding that the core wind speed and core height can be used to diagnose turbulence effects in the region beneath the jet, considering that strong core wind speed are usually associated with strong turbulence below (Bonin et al., 2015) and above the core (Conangla and Cuxart, 2006). In this study, both core wind speed and core height were tested as indicators for the classification of LLJ (not shown here), however, it was not possible to identify groups with consistent patterns in the LLJ characteristics. Hence, we found that the vertical velocity variance ($\sigma_w^2$) is a more meaningful indicator as it effectively describes the link between the jet winds and surface-atmosphere exchanges. This turbulence quantity approximates the vertical component of the TKE during stable conditions (Banta et al., 2006). Here we assume that the $\sigma_w^2$ observations at the first range gate ($238\,\mathrm{m}$ agl) of the DWL at the urban site provides a representative proxy for vertical mixing in the nocturnal urban boundary layer. Each LLJ event is classified according to the mean nocturnal average $\sigma_w^2$ (between sunset and sunrise). Three classes are defined based on thresholds defined by the median and the 75th percentile of the distribution of $\sigma_w^2$ (Fig. 3):



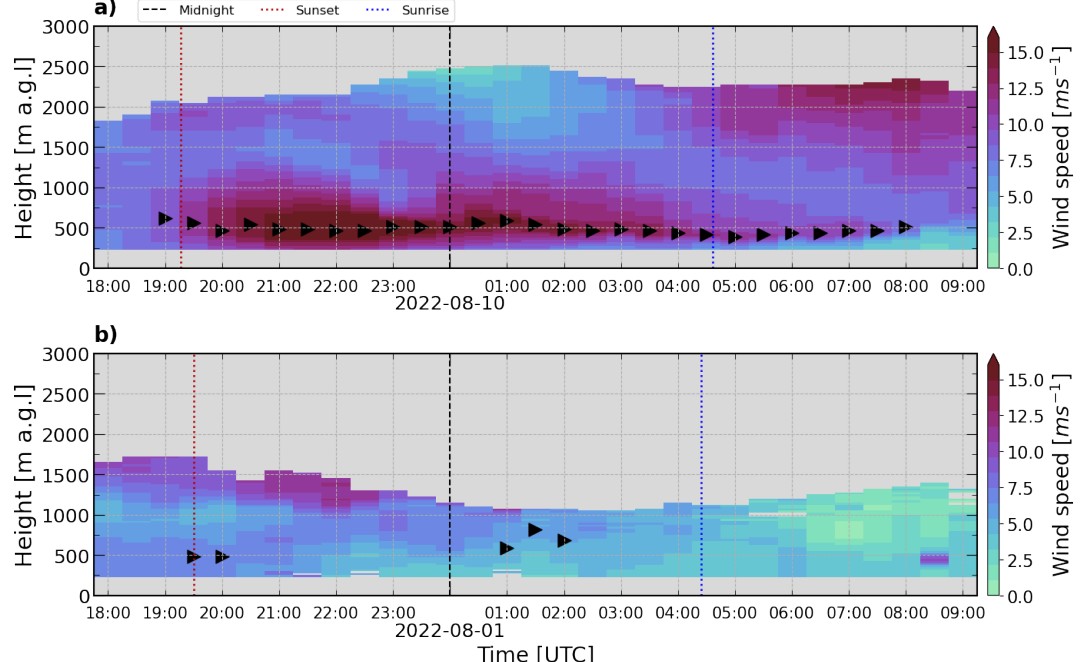

**Figure 2.** Time-height evolution of the horizontal wind speed. The temporal series are composited by 30-min average profiles retrieved by measurements performed by the WindCube Scan 400S at QUALAIR-SU (Urban site) between 18 - 09 UTC on a), between 9 and 10 August 2022, and b) 31 July to 1 August 2022. In both cases, the black triangles represent the core of the Low-Level Jet detected by the automatic procedure.

indicating low ($< 0.11\,\mathrm{m^2s^{-2}}$), intermediate ($0.11\,\mathrm{m^2s^{-2}} \leq \sigma_w^2 < 0.23\,\mathrm{m^2s^{-2}}$), and high ($\geq 0.23\,\mathrm{m^2s^{-2}}$) vertical velocity variance, respectively.

### 3.3 LLJ characteristics

Based on the 79-day period spanning from 15 June 2022 to 31 August 2022, 55 nights (70%) exhibit a LLJ event at the urban QUALAIR-SU site. 96% (53 out of 55 nights) of these LLJ events were also detected over the suburban site at SIRTA, showing that the nocturnal LLJ is a regional phenomenon in the Paris region. 6 out of 55 LLJ events were detected during cloudy conditions, but the complex interactions with cloud dynamics are beyond the scope of this work. Therefore, this study focuses on 49 nights (62% of the total of 79 nights in the study period) with a LLJ event detected at QUALAIR-SU under cloud-free conditions, in which 830 individual 30-min profiles (i.e. $415\,\mathrm{h}$) were associated with a jet. According to these results, it is possible to say that the LLJ activity during summer Paris 2022 is high, specially when compared to previous studies conducted in Paris during the autumn period or to other European studies of summertime LLJ activity. In the local context, Cheliotis et al. (2021) reported a 32% occurrence of LLJ during the fall of 2014 in Paris by using data collected by a DWL installed at QUALAIR-SU site. A climatological study at Cabauw, The Netherlands, found that LLJ occurred  32% of the time during





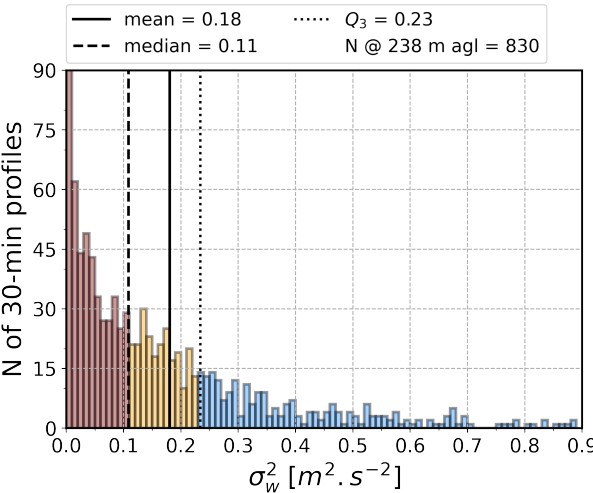

**Figure 3.** Histogram of the variance of the vertical velocity variance ($\sigma_w^2$) retrieved from vertical stare measurements at the first range gate at the urban site (238 m agl). The colors correspond to the $\sigma_w^2$ classes and are : low vertical mixing $\sigma_w^2 < 0.11\,\mathrm{m^2s^{-2}}$ (red), moderate vertical mixing $0.11\,\mathrm{m^2s^{-2}} \leq \sigma_w^2 < 0.23\,\mathrm{m^2s^{-2}}$ (orange), and strong vertical mixing $\sigma_w^2 \geq 0.23\,\mathrm{m^2s^{-2}}$ (blue). N = total number of 30-min profiles and Q$_3$ = Quartile 3, upper quartile.

summer, based on 7 years of SODAR profile data (Baas et al., 2009). At Utö, Finland, another climatological study reported an average occurrence of LLJ of 28% for the month of July, based on 2 years of DWL observations (Tuononen et al., 2017). Outside Europe, a long-term study in Florida, USA, reported a 47% occurrence rate of LLJ during summer, based on 4 years of SODAR data (Karipot et al., 2009), and another work by Duarte et al. (2015) also reported a 47% occurrence rate in a summer 290 study in South Carolina, USA.

During the study period, the LLJ core height ($Z_{LLJ}$) ranges between 238 m agl and 950 m agl (Fig. 4a), i.e., throughout the entire height range considered for detection (Section 2.3). Note that for the 7% of 30-min profiles, when the core height is detected at 238 m agl, the actual LLJ core may at times be located in the instrument blind-zone, i.e. below that level. Most jets (64%) have a core height between 300 and 500 m agl, with 350-400 m agl being the most common interval (22%). The 295 median value of the LLJ core height is 388 m agl. These results are similar to previous case study analyses that found the core height of LLJ in the Paris region at 400 m agl (Klein et al., 2019) and compare well with observations conducted over Greater London that reported core heights between 300 and 400 m agl (Tsiringakis et al., 2022). The analysis carried out at Boulogne-sur-Mer, France (about 200 km north of Paris) reports LLJ core heights < 200 m (Roy et al., 2021). At this coastal site, it is likely that land-sea breezes play a role in the formation of the jet which might explain the occurrence of low-altitude 300 core heights. At other European sites with coastal influence (Cabauw and Utö) lower $Z_{LLJ}$ (140 m agl) were observed during the summer (Baas et al., 2009; Tuononen et al., 2017).

The histogram of the LLJ core wind direction ($WD_{LLJ}$) (Fig. 4b), shows that LLJs are mostly observed under Easterly-Northeasterly flow during the study period in summer 2022. 66% of LLJs have a core wind direction between 0° and 105°,





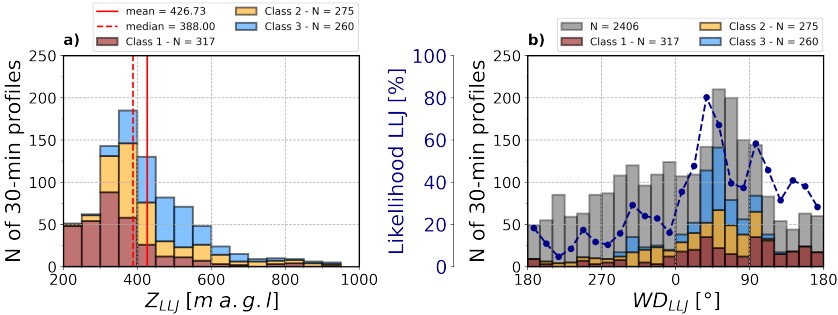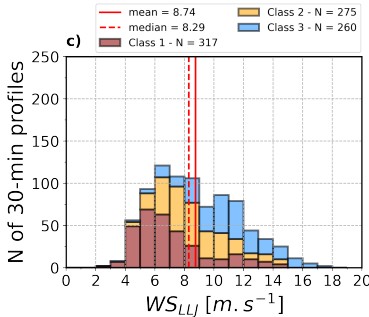

**Figure 4.** Histogram of a) the LLJ core height ($Z_{LLJ}$), b) the LLJ core wind direction ($WD_{LLJ}$), and c) the LLJ core wind speed ($WS_{LLJ}$). The colors of the bars represent the vertical mixing ($\sigma_w^2$) classes as defined in Fig. 3. In b) the gray bars denote the wind direction distribution at $400\,\mathrm{m}$ agl for all nocturnal periods between 15 June and 31 August 2022, and the dashed dark blue line represents the likelihood of a LLJ being detected from a given wind direction.

with $30°$-$60°$ being the most common interval. LLJ profiles with northwesterly flow ($300° < WD_{LLJ} \leq 360°$) present an
occurrence of 12%, while southeasterly flow ($105° < WD_{LLJ} \leq 180°$) contributes $\approx 14\%$ of the observed jets. The remaining
8% of the profiles are distributed in the interval $180°$ to $300°$. To assess how the LLJ over Paris relates to the general regional
flow, the overall nocturnal wind direction distribution at $400\,\mathrm{m}$ agl (i.e. the level of the median $Z_{LLJ}$) is compared to the wind
direction occurrence of LLJ events. The wind direction distribution of the LLJ follows a similar pattern as observed for the wind
during the entire study period, where the northeasterly-easterly flow is predominant (44% of 2375 profiles). The likelihood of
observing a LLJ is also higher in this sector with a 60% probability between $0°$ and $105°$, and the highest occurrence (80%)
in the $30°$-$45°$ interval. Note that only two out of 49 jets ( 4%) present a southwest ($190°$-$300°$) wind direction, a sector that
usually has a much larger frequency during summer nights with respect to the long-term climatology (2006-2022, Section 2.1).

     Fig. 4c show that the LLJ core wind speed ($WS_{LLJ}$) varies between $2\,\mathrm{ms^{-1}}$ and $19\,\mathrm{ms^{-1}}$. For the majority of the jets
(85%), the core wind speed ranges between $4\,\mathrm{ms^{-1}}$ and $12\,\mathrm{ms^{-1}}$, with a median value of $8.3\,\mathrm{ms^{-1}}$. This is again similar
to the modeling results of the two-day case study in Paris reported by Klein et al. (2019), and comparable to a climatological
study conducted in Jülich, Germany, a suburban area located $370\,\mathrm{km}$ from Paris in the northeasterly direction where the median
summer value of $WS_{LLJ}$ is $8.3\,\mathrm{ms^{-1}}$ (Marke et al., 2018). Similarities are also found with Cabauw, The Netherlands (Baas
et al., 2009), which is located in a suburban area $200\,\mathrm{km}$ from Paris in the northeasterly direction where the median value of
$WS_{LLJ}$ for summer is $\approx 9\,\mathrm{ms^{-1}}$. Given that both Cabauw and Jülich are located roughly upwind of Paris under Northeasterly
flow, all these sites could be affected by the same LLJ event if the phenomenon has a large enough spatial extent as shown e.g.
by Klein et al. (2019), which could explain these similarities in LLJ wind speed. However, the question of spatial extent goes
beyond the scope of the current study. The median $WS_{LLJ}$ found in this study is lower than the wind speeds reported for the
summer in Utö, Finland, $11.6\,\mathrm{ms^{-1}}$ (Tuononen et al., 2017), and results from a study in Florida, USA (Karipot et al., 2009).



## 3.4 LLJ signature in the vertical wind profile

As introduced in Section 3.2, a $\sigma_w^2$-based classification system is used to identify LLJ events with common characteristics, and then study their potential impacts on the near-surface atmosphere. Fig. 4 shows that low $\sigma_w^2$ is mostly related to jets with a low core height between 238-350 m agl, which in turn present predominantly east-southeast wind direction (90°-180°), and low to intermediate core wind speed $< 7\,\mathrm{m\,s^{-1}}$. It should be noted, however, that LLJ events with core wind speed $> 10\,\mathrm{m\,s^{-1}}$ are also found in the low $\sigma_w^2$ class. These outliers are later discussed in Section 3.5. Jets with intermediate $\sigma_w^2$ occur mostly in two

wind direction sectors: 300°-345° and 45°-105°. Their core height presents intermediate altitudes between 300 and 450 m agl, and wind speeds between 5-12 $\mathrm{m\,s^{-1}}$. In the strong $\sigma_w^2$ class, the core height tends to be $> 450$ m agl, with a predominant northeasterly wind direction (30°-60°), and wind speeds $> 8.3\,\mathrm{m\,s^{-1}}$.

Fig. 5a presents the mean LLJ horizontal wind speed profile for each $\sigma_w^2$ class. Consistently with the histograms (see Fig. 4), the low $\sigma_w^2$ class present rather low values of wind speed throughout the profile. The core height of this profile is observed at

350 m agl and the wind speed at this altitude is 5 $\mathrm{m\,s^{-1}}$. The minimum wind speed above the core is 3.8 $\mathrm{m\,s^{-1}}$ and occurs at 1000 m agl, above this height the wind speed gradually increases. Similarly, the mean wind profile for the intermediate $\sigma_w^2$ class presents a core height at a slightly greater altitude of 400 m agl, but with a slightly higher wind speed, about 6.2 $\mathrm{m\,s^{-1}}$. The minimum above the core is close to that of the low-variance class. The mean profile associated with the strong $\sigma_w^2$ class exhibits a distinctly different shape and stronger wind speeds. The core height is located at 475 m agl, i.e. 75 m higher than the

moderate $\sigma_w^2$ profile. The core wind speed is about 9.2 $\mathrm{m\,s^{-1}}$, which is 50% and 30% stronger than the core wind speed for the low and intermediate $\sigma_w^2$ class, respectively. The minimum above the core is about 3.8 $\mathrm{m\,s^{-1}}$ and located at 1750 m agl, i.e. at greater altitudes than for the other two classes.

The respective median profiles of $\sigma_w^2$ for each class are presented in Fig. 5b, which illustrate that LLJ core height not only influence the horizontal wind speed but also the vertical mixing. For all three median profiles, the strongest vertical mixing

is retrieved from data recorded at the first range gate of the lidar (238 m agl), i.e. closer to ground level. Above this height, the median $\sigma_w^2$ values gradually decrease, reaching values close to zero, above the respective jet core shown in Fig. 5a. As the minimum recorded altitude of the vertical stare depends on the blind zone of the DWL, no information on the vertical velocity variance below 238 m agl is available. In general, the shape of the three median profiles agrees with those reported in previous studies (Banta et al., 2006; Bonin et al., 2015; Theeuwes et al., 2019), which have found that $\sigma_w^2$ is clearly related to

the turbulent heat fluxes in the near-surface atmosphere, especially in the UBL.

To conclude, the classification system used here in this study allows us to identify the following trends: i) in the low $\sigma_w^2$ class the LLJ profiles are detected in a wind direction sector between 0° and 180°. This class represents all the profiles detected in the southeast sector between 105° and 180°, all the low altitude cases in the interval between 250 and 300 m agl, and is the dominant class for profiles with a core wind speed $< 6\,\mathrm{m\,s^{-1}}$. ii) The intermediate $\sigma_w^2$ class is detected in a wide wind direction

sector between 300°-115°, and dominates the northwest sector between 300°-360°. The values of core height range between 300-450 m agl, while the core wind speed varies between 5-12 $\mathrm{m\,s^{-1}}$. iii) Finally, profiles in the strong $\sigma_w^2$ class occur in the





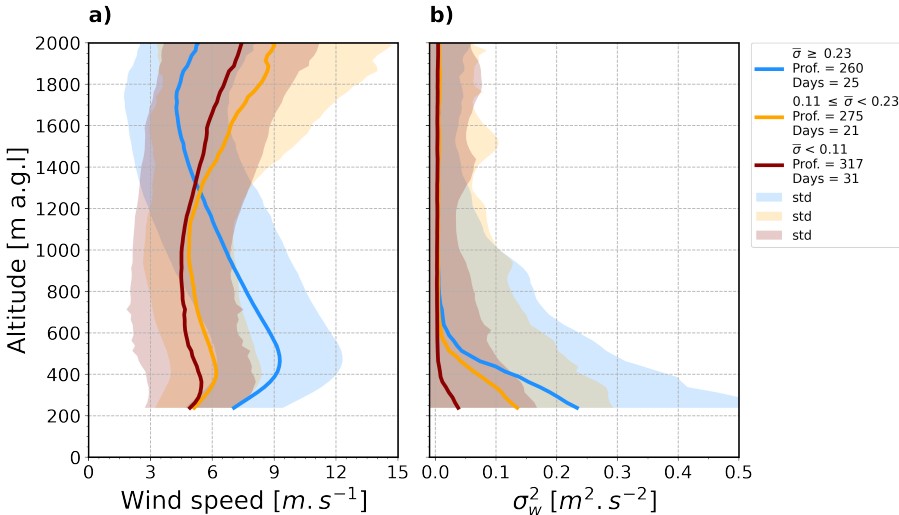

**Figure 5.** Mean profiles of a) horizontal wind speed and b) median profiles of vertical velocity variance $\sigma_w^2$. The solid lines represent the composite profile with all 30-min profiles detected as LLJ, and the shadows denote 1±SD. Each panel presents the respective profiles for each $\sigma_w^2$ class: red is low $\sigma_w^2$, orange is moderate and blue is strong $\sigma_w^2$.

wind direction sector between $0°$-$115°$, but are usually found at the northeast sector between $30°$ and $60°$. LLJs in this category can present core height at high altitudes $> 400\,\mathrm{m}$ agl with a strong core wind speed $> 8.3\,\mathrm{m\,s^{-1}}$.

### 3.5 LLJ nocturnal evolution

According to Stull (1988), a LLJ typically forms during the night and reaches its maximum wind speed before dawn hours, between midnight and 4h local time. Fig. 6a presents an example of a well-defined nocturnal LLJ event observed in Paris during the night between the 16-17 July 2022, showing both the characteristics of the LLJ core and the minimum above. The $WS_{LLJ}$ is about $7.5\,\mathrm{m\,s^{-1}}$ before sunset, then it increases and reaches its maximum ($12.5\,\mathrm{m\,s^{-1}}$) at $2.5\,\mathrm{h}$ after sunset (22h30 UTC and 00h30 LT), then starts to decrease again up from about $1\,\mathrm{h}$ before sunrise. The $Z_{LLJ}$ is about $750\,\mathrm{m}$ agl at sunset when

the convective boundary layer is still collapsing. When $WS_{LLJ}$ starts to increase after sunset (21h UTC), $Z_{LLJ}$ decreases to around $400\,\mathrm{m}$ agl and remains around this altitude throughout the night. At sunset, the jet core follows a northeasterly flow that veers toward the east over the course of the night by $90°$. The lack of sudden changes indicates that there are no changes in air mass.

The development of the wind speed minimum above the LLJ core (see Fig. 5a) and the evolution of its characteristics are

in accordance with the momentum budget of the atmospheric column in the ABL during a LLJ event (Blackadar, 1957). The wind speed is weak ($\approx 2.5\,\mathrm{m\,s^{-1}}$) at sunset and then experiences a slight increase over the course of the night. The height of this minimum is located about $900\,\mathrm{m}$ above the LLJ core at sunset and then gradually decreases until it merges with the jet core at the time of the jet dissipation. The wind direction of the jet core and the minimum above differ by about $50°$ at sunset and




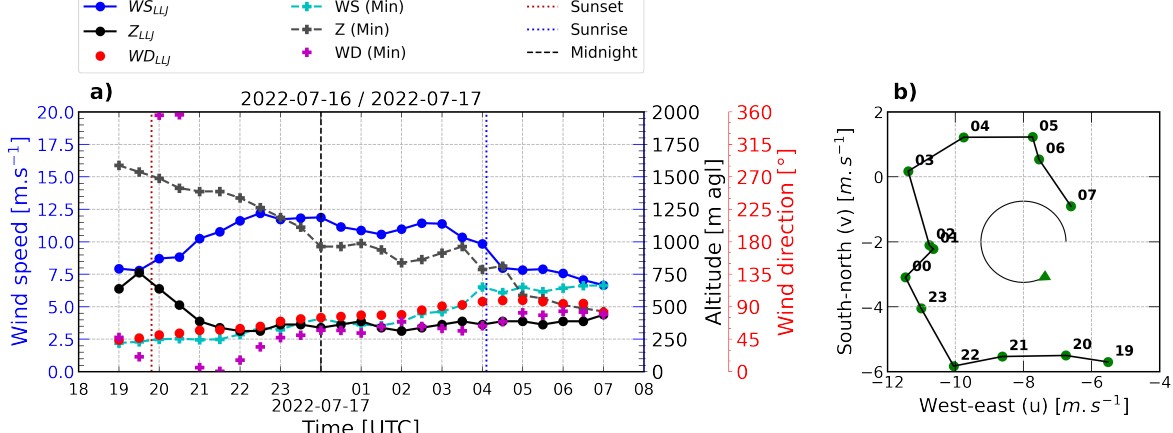

**Figure 6.** Nocturnal evolution of an individual LLJ event (16-17 July 2022) with a) illustrating the jet characteristics: core wind speed ($WS_{LLJ}$) and the minimum above ($WS_{Min}$), core height ($Z_{LLJ}$) and the height of the minimum above ($Z_{Min}$), core wind direction ($WD_{LLJ}$) and wind direction of the minimum above ($WD_{Min}$); and b) the wind hodograph based on the meridional and zonal components of the wind ($u, v$) at the height of the core of the jet. The green arrow indicates the sense in which the wind is veering during the night (clockwise). Labels indicate the hour from 19h of 16 July to 07h of 17 July (UTC).

then slowly converge over the course of the night before converging again at about $1\,\text{h}$ past sunrise. Overall, the wind speed,
wind direction and height in the jet core and the minimum above show distinct contrasts during the time of jet formation and then reach a common point at 7h UTC when the jet is completely dissipated and the momentum is distributed evenly over the entire column in the ABL.

This LLJ evolution (Fig. 6) suggests that the Inertial Oscillation (IO) mechanisms (Blackadar, 1957; van de Wiel et al., 2010) is highly relevant for the jet dynamics in the Paris region. The IO is characterized by an oscillation of the wind above the
elevated nocturnal temperature inversion with a period of $T = 2\pi f$, where $f$ is the Coriolis parameter ($f = 2\Omega \sin\theta$, with $\Omega$ and $\theta$ being the angular speed of the Earth's rotation and latitude, respectively). At the QUALAIR-SU latitude ($\theta = 48.8466$), this gives $T \approx 16\,\text{h}$, which is approximately the duration of the most persistent LLJ events observed here, e.g. 7-10 August (see Fig. 8). The amplitude of this wind speed oscillation is related to the magnitude of the ageostrophic velocity component, and can cause the core wind speed to reach supergeostrophic magnitude. In addition to a gradual increase and decrease (wind speed
amplitude hereafter) of the core wind speed, the IO is associated with a a clockwise change in wind direction, which is clearly evident from the time series (Fig. 6a) and also illustrated by the hodograph of the meridional and zonal wind components ($u, v$) (Fig. 6b).

Fig. 7 presents the nocturnal evolution of the core wind speed for the 49 jets observed under cloud-free conditions during the study period, classified into three groups of $\sigma_w^2$ intensity (section 3.2). In the low $\sigma_w^2$ class, 15 out of 19 (79%) cases that do
not show the wind speed amplitude, as opposed to four visible outliers (16 and 19 July, and 11 and 13 August). These outliers do not show common behavior in terms of duration, acceleration or peak wind speed. These cases are associated with a strong





stable atmospheric stratification caused by the advection of hot air masses (Kotthaus et al., 2023), but these details are beyond the scope of this study and will be investigated elsewhere. Omitting those outliers, the behavior of the median wind speed curve for the low $\sigma_w^2$ class is rather flat with a mean value of $6.23\,\mathrm{m\,s^{-1}}$. For the events with moderate $\sigma_w^2$ values (Fig. 7b), the median wind speed curve is stronger, with a mean value of $8.07\,\mathrm{m\,s^{-1}}$. It increases slightly, starting from $2\,\mathrm{h}$ after sunset and reaches its maximum at about 4-5 h later, before starting to slowly decay until dissipation. Finally, for the strong $\sigma_w^2$ class (Fig. 7c) most of the events exhibit a clear wind speed amplitude in wind speed. The acceleration starts with the jet formation and lasts until $3\,\mathrm{h}$ after sunset. A peak in the core wind speed is observed at around midnight ($\pm 2\,\mathrm{h}$), followed by a slow deceleration until dawn hours with a mean value of $10.2\,\mathrm{m\,s^{-1}}$. The median $WS_{LLJ}$ is $9\,\mathrm{m\,s^{-1}}$ at $1\,\mathrm{h}$ before sunset (i.e., in the order of magnitude as the peak of the wind speed median in the moderate $\sigma_w^2$ class) and then a gradual increase is produced until it reaches the maximum peak at $2\,\mathrm{h}$ after sunset, which lasts about $2\,\mathrm{h}$. The $WS_{LLJ}$ gradually starts to decay up from about $4\,\mathrm{h}$ after sunset until it reaches a minimum right before sunrise. This is explained by the formation of the convective boundary layer during the morning.

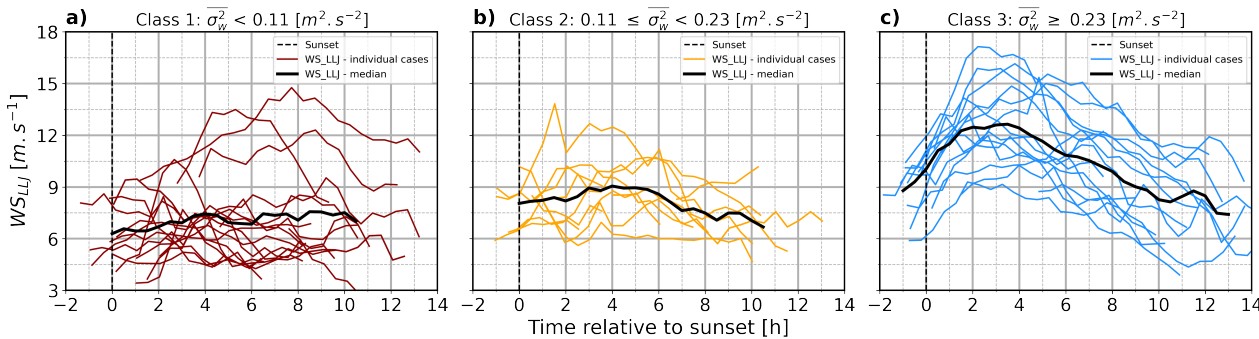

**Figure 7.** The evolution of the core wind speed throughout the night for each LLJ event classified by a) low, b) intermediate, and c) strong vertical mixing. The black line represents the median for each group, which is calculated for times with a minimum of three samples.

## 3.6 Temporal distribution of the LLJ characteristics

Fig. 8 presents the time series of the nocturnal evolution for the three jet core characteristics (height, wind speed and wind direction) for each event during the study period, including the cloudy nights. An amplitude in wind speed can clearly be identified for 25 out of 49 (i.e. 51%) jets, mainly observed for jets with strong vertical velocity variance (as shown in Fig. 7c) and high core height. This trend is also valid for short-duration cases ($< 4\,\mathrm{h}$), for example, 16 July. Nocturnal evolution of the core direction (Fig. 8b) shows that 28 out of 49 (i.e. 57%) of jets present a persistent clockwise veering with an average change in wind direction of $3.7\,^\circ\mathrm{h^{-1}}$. Two cases particularly stand out with a strong veering ($> 5\,^\circ\mathrm{h^{-1}}$): 28 June and 21 August. They fall into the intermediate and low $\sigma_w^2$ class, respectively, and both emerge from a northwesterly direction ($322^\circ$ and $282^\circ$, respective initial core direction). As discussed (Section 3.5), both the wind speed amplitude and the clockwise veering are signatures of the IO formation mechanism. However, not all cases with wind speed amplitude (Fig. 6a) present necessarily a



strong clockwise veering (Fig. 6b). For example, jets detected between the 7 and 9 of August show the wind speed amplitude,
but their change in core wind direction is less than $1\,^{\circ}\,\mathrm{h}^{-1}$, remaining almost constant in wind direction.

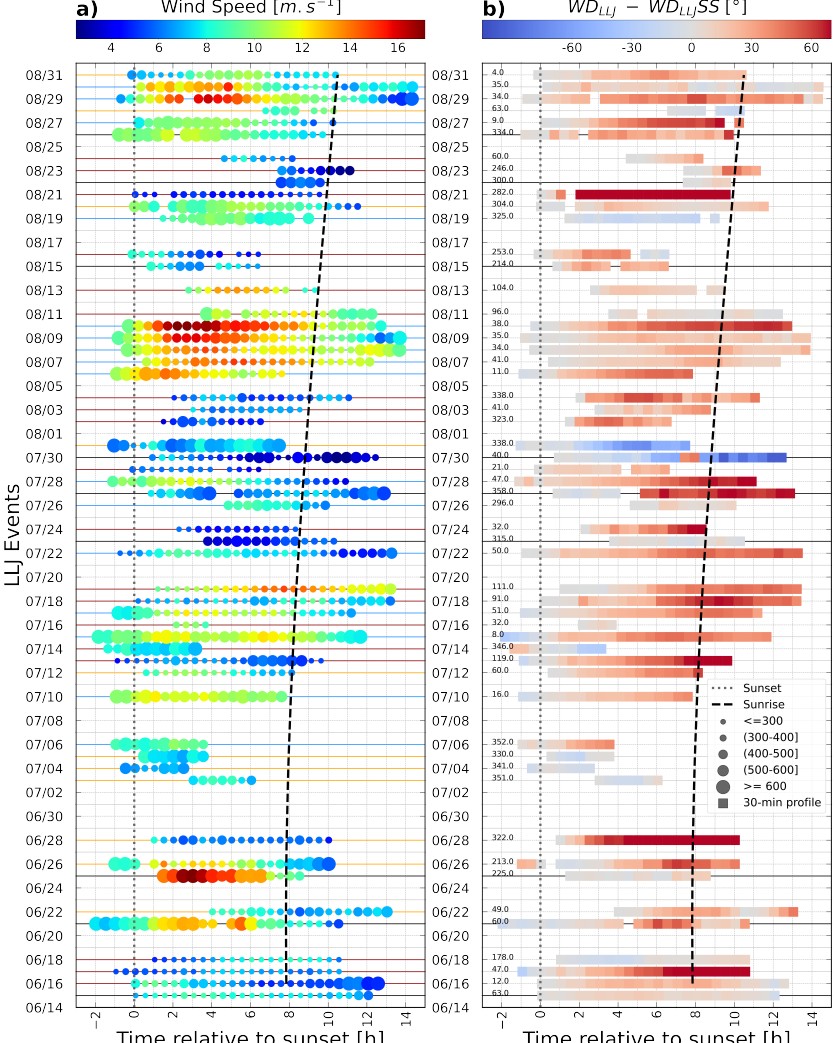

**Figure 8.** Nocturnal evolution of the LLJ events during the study period 15 June - 31 August 2022 at 30 min resolution in time relative to sunset (dotted line). In a) dot size and color indicate the jet core altitude ($Z_{LLJ}$) and wind speed ($WS_{LLJ}$), respectively. Horizontal lines represent the vertical mixing category of each case: low (red), intermediate (orange), and strong vertical mixing (blue). b) Core wind direction ($WD_{LLJ}$) variability relative to the wind direction at sunset or initial wind direction when the jet is formed in the middle of the night. Horizontal black lines mark dates that are excluded from further analysis due to cloud conditions in a) and b). The dashed line marks the time of sunrise.

These time series showcase in detail the variability in the LLJ duration as well as the timing of both formation and dissipating. Results show that 29 out of 49 (i.e. 59%) jets last almost the entire night, emerging at sunset $\pm 2\,\mathrm{h}$ and dissipating at sunrise



$\pm 2$ h. However, jets can emerge in the middle of the night and continue even after sunrise. 31 out of 49 (i.e. 61%) cases last more than 8 h, including cases with late starting time (up to 4 h after sunset). During the study period, the maximum LLJ duration is 15 h, the minimum duration is 2 h (by conceptual definition), and the mean duration is 9.5 h.

The day-to-day variability highlighted in Fig. 8a reveals the influence of synoptic conditions for the LLJ characteristics. While some cases appear to be isolated (e.g. 28 June and, 10 July), during consecutive LLJ detection the characteristics of their core may vary night after night or may be similar each night. Ten events are detected between 10-19 July (heatwave period) with strong day-to-day variability in terms of $\sigma_w^2$ levels and the core characteristics. During this period, jets at high core height and strong wind speeds are associated with strong $\sigma_w^2$, while slow and low-altitude jets present weak $\sigma_w^2$. These cases present differences in terms of nocturnal evolution and duration regarding the 11 and 13 August, which are of easterly origin and combine relatively strong wind speeds with low core height and low vertical velocity variance (outliers in Fig. 7).

The persistence of synoptic conditions results in a certain clustering of the LLJ events. For example between 15 and 18 June, four consecutive events present similar duration (10-12 h), with similar speeds (6-9 m s$^{-1}$), and similar core heights at altitudes ($< 400$ m agl). In the period between 6 and 10 August, the detected events present similar core characteristics with strong wind speed amplitudes at high altitudes, occurring in the northeast flow with a wind direction veering becoming more pronounced from one night to the next. These cases present very similar nocturnal evolution and duration, and all of them fall into the same $\sigma_w^2$ class (strong vertical mixing). In the period between 3 and 6 July again, after four consecutive days with no detection, four short events ($< 5$ h) are found at medium to high altitudes between 400 and 600 m agl, with wind speeds between 10 and 12 m s$^{-1}$.

As discussed in Section 3.3, LLJ in a narrow easterly sector ($80°$-$115°$) are all associated with low vertical velocity variance. Fig. 8 reveals that these specific LLJ events (18 and 19 July, and 11 and 13 August) in fact present the high wind speed outliers detected in Section 3.5 with a distinct wind-speed amplitude (Figure 7a) and clockwise veering (Fig. 8b). It is further evident that these easterly jets are found at relatively low altitudes ($< 350$ m agl) suggesting a channeling along the low topography of the river Marne basin towards the East of Paris could play a role in the acceleration mechanisms. These processes will be investigated in future studies.

### 3.7 LLJ impacts on UHI

Cloudiness and wind are environmental variables that exert an essential control on the intensity of the UHI and its nocturnal evolution. The UHI of the canopy layer is generally strongest during cloud-free nights and low wind speed conditions, following days with the same characteristics (Oke et al., 2017). Fig 9a presents the relationship between the nocturnal mean surface wind speed at the rural site (yellow dot in Fig. 1) and the nocturnal mean UHI intensity in the Paris region, including only cloud-free nights after fair-weather days to focus the analysis on wind effects only. The UHI intensity decreases exponentially with the increasing wind speed. Based on observations from different cities in Canada, Oke (1973) proposed an empirical formulation where the mean $\Delta$UHI is proportional to the mean nocturnal surface wind speed ($\Delta UHI \propto \overline{ws}^{-\kappa}$, being $\kappa$ a dimensionless parameter often equal to 0.5). A best fit to the Paris data following this relation is included in Fig. 9a and describes a function



that follows the drop in UHI due to the increase in wind speed, with an asymptotic shape defined by the threshold UHI < ∼1 °C. For several nights in this study, the ΔUHI deviates quite clearly from this best fit, this was also the case in Oke (1973).

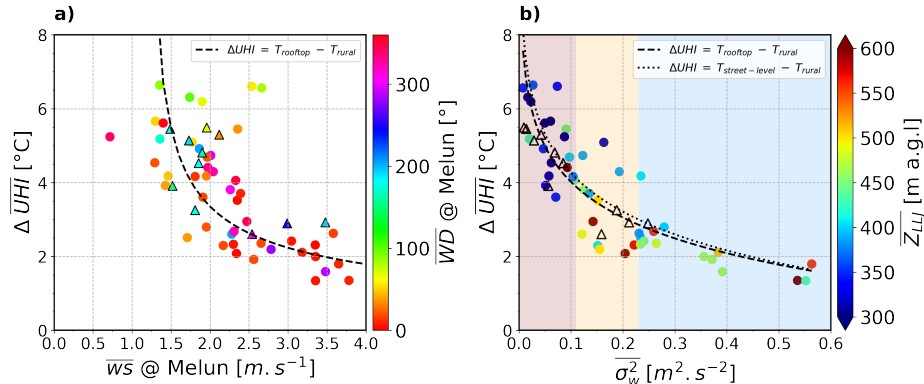

**Figure 9.** Relations between nocturnal average of ΔUHI intensity for all cloud-free nights in the study period and a) 10 m agl wind speed at Melun, the rural reference site (see yellow dot in Fig. 1). The dots are colored by surface wind direction at Melun. The dashed black curve is the best fit to the data: $y = 3((x - 1.2)^{-0.5})$, and it follows the empirical relationship described by Oke (1973) b) the vertical velocity variance ($\sigma_w^2$) at 238 m agl, dots are colored by the LLJ core height above the ground surface. The dashed black curve represent the best non-linear fit found for the data collected in the Paris region, and determining ΔUHI intensity with QUALAIR-SU data. The dotted black curve represents the same but using data collected by a surface-based station installed at Boulevard de Capucines for a shorter time period (see Section 2.4). Both curves represent the equation: $y = -1.39 \log(x) + 0.84$. The background shading indicates the LLJ classes described in Section 3.2. In all subplots, dots and triangles represent nights with and without LLJ events, respectively.

For the current study period, a certain rural surface wind speed (e.g. $2.5\,\mathrm{m\,s^{-1}}$) is associated with a large range of ΔUHI intensities, meaning that surface winds do not appear to be a very accurate predictor. In fact, the rural surface wind speed

shows a rather large spread in relation to urban boundary layer $\sigma_w^2$ (See Fig B1) presumably because the latter responds to both mechanical and buoyancy driven turbulence. Fig 9b shows that ΔUHI has a more pronounced relation with vertical velocity variance. This is consistent with findings by Bonin et al. (2015) and Banta et al. (2006), who showed that the mixing below the LLJ core can affect the turbulent exchange processes near the surface. Table 2 presents a summary of the average values of ΔUHI and the LLJ characteristics, corresponding to each $\sigma_w^2$ class. The ΔUHI is proportional to the vertical mixing

($UHI \propto \sigma_w^2$) and the vertical mixing is proportional to the core height of the jet ($\sigma_w^2 \propto Z_{LLJ}$). Previous studies have shown that jets at low altitudes (up to $300\,\mathrm{m}$ agl) tend to present low to moderate core wind speeds (Banta et al., 2002; Karipot et al., 2009; Carroll et al., 2019), so that weak wind shear only produces little mechanical mixing. These low-altitude jets are often associated with a strong temperature inversion over the rural surface, so that the stable stratification of the atmosphere leads to a more effective decoupling from the surface friction. For these conditions of strong atmospheric stratification and weak

momentum transport by the LLJ, the cool air remains locally over the vegetated rural surfaces where it was generated through efficient radiative cooling during the cloud-free nights. Both advection and vertical mixing processes are weak and hence support the formation of very strong ΔUHI (Hu et al., 2013; Lin et al., 2022). Therefore, one can say that the surface wind



speed relation (as described e.g. by Oke (1973)) may not sufficiently consider the vertical mixing effects. Here we demonstrate that $\sigma_w^2$, obtained commonly at one measurement location at a height above the roughness sublayer but below the LLJ core, can

be considered a representative indicator of the mixing in the nocturnal urban boundary layer and hence a powerful predictor for the $\Delta$UHI intensity.

The consistency of the relationship between $\Delta$UHI and $\sigma_w^2$ implies that, according to the data presented in this study, $\sigma_w^2$ is a better predictor for $\Delta$UHI than the surface wind speed. The curves in Fig 9b represents the best non-linear model fitted to the data collected during the nights with a LLJ event, using the $T_{air}$ data collected at QUALAIR-SU and by data collected by

475 the IOP station located at Boulevard des Capucines (see Section Section 2.4). The similarity between the two curves indicates that the relationship between $\Delta$UHI and $\sigma_w^2$ is preserved even when using QUALAIR-SU data collected at roof-level. Table 3 presents a summary of the errors in $\Delta$UHI prediction when using rural surface wind or urban $\sigma_w^2$, respectively, as a predictor.

Note that the relation between $\Delta$UHI and $\sigma_w^2$ remains valid even during nights when no consistent LLJ was detected by the automatic procedure (black triangles in Fig 9b), and other synoptic processes (such as fronts) are likely to modulate the

480 atmospheric mixing. 53% of the jet-free nights fall into the low $\sigma_w^2$ class (red shading), and all show $\Delta$UHI > 3.5 °C. Future studies are going to investigate why there is no LLJ detected for some nights although a low $\sigma_w^2$ indicates the atmospheric stratification that is likely to favor decoupling.

**Table 2.** Mean values of core height ($Z_{LLJ}$), core wind speed ($WS_{LLJ}$) and the corresponding $\Delta$UHI for each class of vertical velocity variance ($\sigma_w^2$).

| $\sigma_w^2$ (m²s⁻²) | < 0.11 | $0.11 \leq \sigma_w^2 < 0.23$ | $\geq 0.23$ |
|---|---|---|---|
| $WS_{LLJ}$ (ms⁻¹) | 6.2 | 8.0 | 10.1 |
| $Z_{LLJ}$ (m agl) | 386 | 490 | 540 |
| $\Delta$UHI (°C) | 5.39 | 3.05 | 1.80 |



**Table 3.** Error metrics for prediction of $\Delta$UHI prediction. In order of appearance from left to right: following the surface wind speed relationship proposed by Oke (1973), the best-fit model to rural surface wind speed at Melun site and vertical velocity variance ($\sigma_w^2$) in the urban boundary layer above the roughness sublayer, respectively. The metrics are Mean Square Error (MSE), Root Mean Square Error (RMSE), Mean Absolute Error (MAE), and determination coefficient ($R^2$).

| $\Delta$**UHI Estimator parameter** | $(3(\overline{ws} - 1.2)^{-0.5})$ | $\sigma_w^2$ |
|---|---|---|
| MSE | 2.89 | 0.65 |
| RMSE | 1.70 | 0.81 |
| MAE | 1.19 | 0.63 |
| $R^2$ | -0.21 | 0.70 |

## 4 Conclusions

For the first time, this study reports a full description of the summertime Low-Level Jet (LLJ) characteristics over the Paris city center. The automatic LLJ detection is based on an existing algorithm (Tuononen et al., 2017), which was tailored to the data set and study area. This algorithm uses horizontal wind profiles at a resolution of 30-min obtained from continuous measurements performed by a high-power scanning Doppler Wind Lidar (DWL), installed on the roof of a tall tower in the city center, at the QUALAIR-SU supersite. LLJs were observed on 55 nights (70%) during the study period between 15 June and 31 August of 2022. Further supported by wind profile observations performed by a second DWL at a suburban site (SIRTA), it was shown that the LLJ is a regional phenomenon: 90% of the LLJ events are detected at both sites. This study is focused on jets during cloud-free conditions, and hence six cloudy nights were excluded from the analysis leaving 49 cases for detailed investigation. Daytime jets were rarely observed during the study period. Thus, this work focuses on nocturnal jets observed between 18h and 9h UTC (20h and 11h local time).

The high temporal and spatial resolution of the DWL measurements allows us to assess the LLJ occurrence and the nocturnal evolution of the characteristics of the jet core (i.e. the wind speed maximum in the vertical profile) such as that height, wind speed, and wind direction. Results show a mean core height of $428\,\mathrm{m}$ agl, with most of the jets ranging between 300 and $500\,\mathrm{m}$ agl. The predominant LLJ core wind direction is the northeast flow ($30°$-$60°$). Very few LLJs were detected with a core wind direction between $190°$ and $300°$. At night, during the summertime period of this study, the prevailing wind directions range between $170°$ and $45°$. Easterly and southeasterly directions are less frequent. The summer of 2022 was exceptional with a predominant flow from the easterly and northeasterly directions. The mean value of core wind speed is $8.9\,\mathrm{m\,s^{-1}}$, but jets can have speeds between 4 and $12\,\mathrm{m\,s^{-1}}$.

To better understand the LLJ characteristics and their potential impacts on the near-surface atmosphere, a classification system was implemented based on the vertical velocity variance-$\sigma_w^2$ (a direct measure of vertical mixing). $\sigma_w^2$ values obtained at $238\,\mathrm{m}$ agl in the urban boundary layer are used to classify the jet events. Three LLJ classes are established according to low





($\sigma_w^2 < 0.11 m^2 s^{-2}$), intermediate ($0.11 m^2 s^{-2} \leq \sigma_w^2 < 0.23 m^2 s^{-2}$), and strong vertical mixing ($\sigma_w^2 \geq 0.23 m^2 s^{-2}$). Jets with low $\sigma_w^2$ are found from $0°$ to $180°$. They constitute the majority of cases of the southeast sector ($105°$-$180°$), most commonly with low altitude (250-300 m agl) and low to intermediate wind speed $< 6\,\mathrm{m\,s^{-1}}$. The intermediate $\sigma_w^2$ class is found for wind directions between $300°$ and $115°$ and dominates in the northwest sector ($300°$-$360°$). The LLJs that fall into this class present a core height between 300-450 m agl, and core wind speeds between 5-8.3 $\mathrm{m\,s^{-1}}$. The strong mixing class is found in a wind

direction sector between $0°$ and $115°$, but most cases are detected in a narrow sector ($30°$-$60°$) at high altitudes $> 400$ m agl and with strong wind speeds $> 8.5\,\mathrm{m\,s^{-1}}$. The results described in this work are consistent with previous LLJ studies at European inland sites (Netherlands, Northern France, Germany) as strong jets tend to be located at higher altitudes. The profile data analyzed in this study are limited to LLJ detection down to altitudes $\geq 238$ m agl. Future studies based on novel scanning strategies are required to assess the frequency and characteristics of the LLJ closer to the ground.

The majority of the LLJs detected during the study period are of long-duration, emerging at about sunset and dissipating at about sunrise. Some cases can emerge in the middle of the night and persist even after sunrise. The mean LLJ duration is $9.5\,\mathrm{h}$, the maximum is $15\,\mathrm{h}$ and the minimum duration is $2\,\mathrm{h}$ (by definition). Besides the LLJ duration, the high temporal resolution and continuous operation of the DWL can provide remarkable insights regarding the nocturnal evolution of the studied jets. For half of the cases (i.e. 51%) the core wind speed changes with a certain amplitude over the course of the night with a peak

at $2\,\mathrm{h}$ around midnight. This trend remains even for short-duration cases ($< 4\,\mathrm{h}$). Additionally, a clockwise veering in the core wind direction is observed for 57% of the jets, with an average change in wind direction of $3.7\,°\,\mathrm{h^{-1}}$. Two special cases were detected with a strong veering ($> 5\,°\,\mathrm{h^{-1}}$), falling into the low and intermediate $\sigma_w^2$ class. These two cases are the only ones that emerge in the northwest wind direction sector and then are veering eastward. They present a low core wind speed $< 6\,\mathrm{m\,s^{-1}}$, and low core height $< 400$ m agl.

The presence of both the signature of the core wind speed acceleration and the core wind direction veering indicate that the LLJs observed in the Paris regions are forming through the mechanism of Inertial Oscillation (IO). The acceleration signature is found for most cases with strong $\sigma_w^2$ and medium to high wind speeds, while it hardly appears for weak LLJ with low $\sigma_w^2$. The wind direction veering appears in all vertical mixing classes but it is most frequent for LLJ with strong $\sigma_w^2$. Four outliers in the low $\sigma_w^2$ class present the acceleration signature with strong wind speed and a late starting time. These cases with strong

core wind speed and low $\sigma_w^2$ are likely to be generated during very stable near-surface atmosphere conditions. In addition, particularly low surface drag may favor the acceleration of these jets as they pass over a basin area of low topography in the East of Paris before reaching the study site. The role of IO in the Paris LLJ formation and the impact of topography require further investigation.

Another important phenomenon that is particularly pronounced under nocturnal stable stratification over rural areas is the

canopy layer Urban Heat Island (UHI) effect. During the study period, and accounting only for cloud-free nights as done for the LLJ assessment, a mean $\Delta$UHI of $3.7\,°\mathrm{C}$ was found with maxima of up to approximately $8\,°\mathrm{C}$. Usually, the surface wind speeds are considered as a key parameter to explain variations in the $\Delta$UHI intensity, as they account for dynamic effects (mostly advection). Here we found that the vertical mixing indicator of $\sigma_w^2$ in the urban boundary layer can be a better predictor for the $\Delta$UHI intensity, as the Mean Absolute Error (MAE) is smaller ($0.63\,°\mathrm{C}$) compared to the uncertainty when using rural





surface wind speed to explain the $\Delta$UHI variations (MAE = $1.19\,°$C). Furthermore, the LLJ core wind speed is less suitable for understanding UHI variations, which is highlighted by four outlier events (16 and 19 July, and 11 and 13 August) with strong core wind speeds and weak $\sigma_w^2$ that are among the nights with the strongest $\Delta$UHI ($\sim 8\,°$C nocturnal average). The vertical turbulence enhances the weakening of the temperature inversion over the city and this is valid for all cloud-free nights, with and without LLJ presence. Different sources of turbulence in the cloud-free atmospheric boundary layer other than the LLJ

hence play a role for the UHI development. Nights characterized by low $\sigma_w^2$ jet events present an average $\Delta$UHI of $5.39\,°$C; while the $\Delta$UHI is lower for intermediate $\sigma_w^2$ ($3.5\,°$C) and minimal for strong $\sigma_w^2$ cases ($1.80\,°$C). The $\sigma_w^2$ classification system is an important contribution of this work as it not only proves to be valuable for the characterization of the range of nocturnal evolution and general characteristics of the Paris LLJ events but also reveals the importance of vertical mixing ($\sigma_w^2$) for the intensity of the UHI.

This study shows that the regional scale synoptic flow (LLJ) has clear implications for the conditions in the urban boundary layer down to the surface. The mechanical turbulence driven by the LLJ clearly affects spatial contrasts in air temperature and is likely to also affect the dispersion of atmospheric pollutants. In the summer of 2022, the LLJ was a very frequent phenomenon with variable characteristics that often suggest that inertial oscillation plays a role in the LLJ formation. A better understanding of the LLJ formation in response to synoptic pressure patterns and topography is required to fully grasp its interactions with

the urban atmosphere.

*Code and data availability.* The data sets will be available in the AERIS-PANAME database (https://paname.aeris-data.fr/) and the code will be available upon request.

**Appendix A: UHI determination**

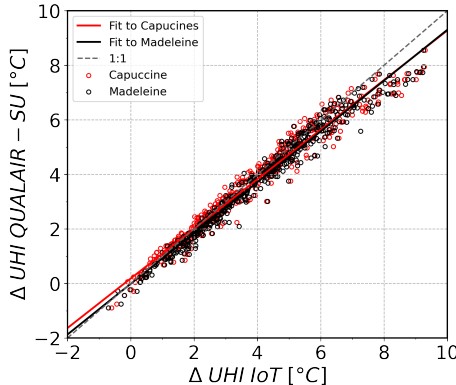

**Figure A1.** Comparison of $\Delta$UHI = $T_{urban}$ - $T_{rural}$ calculated using a rooftop site at QUALAIR-SU ($20\,$m agl) against $\Delta$UHI determined for two street-level: Boulevard des Capucines and Place de la Madeleine. Solid lines represent the best linear fit for each group of data.



**Appendix B: LLJ impacts on UHI**

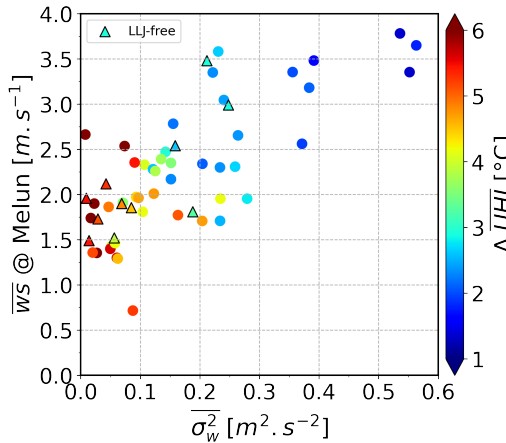

**Figure B1.** Nocturnal average of $10\,\mathrm{m}$ agl wind speed at Melun (rural site) as function of the nocturnal average vertical velocity variance ($\sigma_w^2$) at $238\,\mathrm{m}$ agl in the urban boundary layer, for all cloud-free nights in the study period. Data are colored by the the nocturnal average of $\Delta$UHI. Dots represent nights with LLJ and the triangles the LLJ-free events.

*Author contributions.*  JC, SK, LT, and MH worked on the conceptualization of this work. JC, MH, and JCD performed the WindCube scan 400s deployment and ensured continuous operation. JC, SK, LT, and MH designed the experiment and JC performed the measurements. AF performed the measurements and data curation of the WindCube scan WLS70. JC, SK, JP, and MAD worked on the computational algorithm implementation and data curation. JC, SK, JP, CT, LT, and MH conducted the investigation. JC writing of original draft. SK, CT, JP, LT, and MH reviewed and edited the manuscript. MH, SK, and LT funding acquisition.

*Competing interests.*  The authors declare that they have no conflict of interest.

*Acknowledgements.*  The authors thank OBS4CLIM ANR France 2030, DIM QI2 and ACTRIS-FR for financing the investment of the Doppler Wind Lidar (DWL) Vaisala WindCube Scan 400s. We thank the QUALAIR-SU platform for hosting the 400s and for the meteorological data, especially to Camille Viatte and Cristelle Cailteau-Fischbach. The authors are grateful to AERIS for providing SIRTA ReOBS data and support for PANAME, as well as to Météo-France for the meteorological surface-based data set. Thanks to the SIRTA team, especially
to Christophe Boitel for his technical support and to Jean-François Ribaud for support on the ReOBS data processing. Authors thanks to EDF R&D/CEREA for data providing data collected by the DWL WLS70. Jonnathan Céspedes is grateful to the Region île de France and VAISALA France SAS for funding the doctoral scholarship. This work is conducted in the context of the COST Action CA18235 PROBE, supported by COST (European Cooperation in Science and Technology, https://www.cost.eu/).



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
