# Peer review of "The Paris Low-Level Jet During PANAME 2022 and its Impact on the Summertime Urban Heat Island"

_EGUsphere, 2024_

## Referee Comment (RC2)

**Review of egusphere-2024-520 "The Paris low-level jet during PANAME 2022 and its impact on the summertime urban heat island"**

**Submitted to Atmospheric Chemistry and Physics.**

This manuscript uses Doppler wind lidar measurements from two sites (one urban and one suburban) for the period 15 June – 31 August 2022 to analyse the summertime characteristics of the nocturnal low-level jet over Paris. The characteristics of the jets identified are similar to previous studies of low-level jets over Europe. Using surface wind and temperature observations, the study also investigates the relationship between the jet and the urban heat island. Vertical velocity variance over the city at 238m agl from the lidar is found to be a better predictor of the urban heat island than 10m winds at nearby rural site. There is not necessarily a relationship between jet strength and the urban heat island with some strong jet nights leading to strong urban heat islands. This is undoubtedly an interesting dataset and the continuous nature of the observations allows a study of the jet evolution which is not usually possible with less frequent radiosonde observations. However, I have some questions outlined below which need considering before the manuscript can be published.

**Major comments**

1) I did sometimes struggle to see what the novelty was in the results – the behaviour of the LLJ seems broadly in line with what would be expected based on other studies at similar latitudes / locations. The novelty seems to be around the interaction of the LLJ and urban environment, but the results seem to indicate the LLJ is not strongly affected by the urban land surface. The main result on the UHI seems to be the link with $\sigma_w^2$ which is irrespective of the LLJ. It would be good to focus on highlighting the novelty in a revised manuscript and particularly why the focus on the LLJ rather than UHI more generally.

2) Much of the analysis focuses on the different classes of $\sigma_w^2$ however these are somewhat arbitrarily determined based on previous research. Can you justify the choice of thresholds? Does this depend on the location / the height at which the measurements are made? Is there a more objective way of determining what they should be?

**Minor comments**

1) p9, line 219. I do not understand what the difference in core time < 15 h means. Just above you stated that a LLJ event must have a LLJ detected in at least 4 consecutive profiles. Does this not mean the time between consecutive LLJ detections has to be 30 mins anyway? Please clarify.

2) p10, line 270. You have chosen to use the first range gate (238m agl) to calculate $\sigma_w^2$. I have sometimes seen issues with data at the first range gate being a bit lower quality / noisier than further up. Did you see any issues with the data at the first range gate which might affect the results? Or to put it another way, are the results very dependent on whether you use the first gate, or say the 2$^{nd}$ / 3$^{rd}$ gate? In a sense this is a slightly arbitrary choice of height at which to calculate $\sigma_w^2$, so it would be good to know the results are not too sensitive to this choice.

3) p12, line 291. Here, and throughout the manuscript, you give very precise values for the heights of measurements (in this case 238m). In reality, given the range gates of the lidar are 75m, it is hard to justify this level of accuracy.

4) p13, Figure 4. The figure mentions classes 1, 2 and 3. These are not actually defined in the text. I assume these correspond to the low, medium and high vertical mixing classes?

5) p13, line 323. I wonder why you don't also give a value of the wind speeds reported in the Karipot et al (2009) study? It's hard to compare meaningfully with the current results otherwise.

6) p14, line 344. "influence" -> "influences"

7) p14, lines 348-350. I wonder if this sentence needs some qualification? I agree $\sigma_w^2$ is related to the surface layer turbulent heat fluxes in a convective boundary layer. I'm not sure this is necessarily the case in a stable boundary layer. It is also certainly true that shear generated turbulence can lead to vertical velocity variance and so surface layer turbulent heat fluxes are not solely responsible for $\sigma_w^2$.

8) p16, line 379. "the IO mechanism is highly relevant .. in the Paris region". This does seem to be the case for the example you look at, but since you have deliberately chosen an example without any other synoptic forcing this is perhaps not surprising?

9) p16, line 380. "$T = 2\pi\,f$" should be "$T = 2\,\pi\,/\,f$"

10) p16, line 381. "$\theta=48.8466$". Needs units (degrees). It also seems to be an unjustifiably precise number, particularly when you then give the period T as ~16 hours.

11) p16, line 384-385. "wind speed amplitude". I find this a confusing phrase. You really mean the amplitude of the variations / oscillations in wind speed. Wind speed amplitude could just be the wind speed itself. I would suggest rephrasing to make sure your meaning is clear.

12) p16, line 385. "with a  clockwise change"

13) p16, line 389. "cases  do not show"

14) p17, line 401. "decay  from about"

15) p17, lines 402-403. This sentence is misleading. As written, it implies that the CBL during the morning is responsible for the gradual decay in jet core wind speed overnight. This cannot be true. What I think you mean is that the onset of the CBL in the morning leads to the jet core wind speed starting to increase again and hence is responsible for the minimum wind speed being right before sunrise.

16) p18, Figure 8b. I assume the number at the start of the line is a wind direction, but the caption doesn't actually say what. Is it the wind direction at sunset or at jet formation? I assume this is what $WD_{LLJ}$ SS means in the title? Again, not actually defined anywhere.

17) p19, lines 425-427. This sentence is confusing "These cases present differences .. regarding the 11 and 13 August…" Which cases? Differences compared to what? Are you trying to say that the 11 and 13 August are different to the other cases?

18) p20, line 460. This sentence is wrong. You have written that UHI is proportional to $\sigma_w^2$ and $\sigma_w^2$ is proportional to $Z_{LLJ}$. In the first case, the relationship is definitely not linear according to figure 9b, and in the second case it also does not appear to be true (though it's harder to judge from looking at the colour of the dots in figure 9b. I agree there is a relationship between these variables, but they are not proportional.

19) p21, line 475. "see  Section 2.4)"

20) p22, line 495. "such as  height, …"

---

## Author Comment (AC1)

This manuscript describes a summertime period of nocturnal low-level jets (LLJs), observed with Doppler wind lidars, and the LLJ impact on the Urban Heat Island (UHI) intensity. The main finding of the study is that the LLJ vertical velocity variance observed at the urban site shows a clearer relation to the UHI intensity than the LLJ wind speed. The study is well-written and includes a very detailed description of methods, the observed LLJ characteristics and the conclusions. The abstract provides a clear and concise overview of the study. I have a few minor suggestions for improvement outlined below.

We thank the referee for reviewing our paper, as well as for the positive point of view and relevant feedback. Please, find our response to your comments point-by-point. Modifications on the manuscript are highlightes with blue text and here in this document we included the page and number line of the track changes manuscript.

General comments

1. The study concludes that the wind speed variance shows a clearer relation to the UHI intensity than the wind speed. I would like to make sure that the comparison is not impacted by the measurement location (near-surface wind speed at the rural site vs. wind speed variance at 238m agl at the urban site). Does the lowest-level wind speed derived from the urban site reveal the same trend (and is therefore more comparable to the variance)? Or, the other way around, are there near-surface wind speed variance measurements available from the rural site?

According to Oke et al., 2017, the UHI intensity is proportional to the regional near-surface wind speed during cloud-free conditions. Hence, we initially selected the wind speed at the rural site of Melun as the reference. However, consistent with your remark, indeed fewer outliers are present when choosing surface wind speed at an urban site (Montsouris Park) instead (see figure below). We have updated Figure 9 now using the central urban wind measurements instead and the Table 3, in which we include winds at both measurement sites. We also implemented multiple adaptations in the the manuscript text accordingly:

**Changes in Section 2.4**

p10, line 244: "Additionally, wind data collected from a 25 m meteorological tower installed at Montsouris Park are used in this study to assess the relationship between ∆UHI and the near-surface wind speed."

**Changes in Section 3.7**

p20, line 481: "The urban wind direction at Montsouris suggests that weak values of ∆UHI are mostly found under prevailing northeasterly flow. This wind direction sector is characterized by LLJ with strong wind speeds and high $\sigma^2_w$ values."

p20, line 488: "The near-surface urban wind speed, sampled at a height (25 m agl) is slightly above the mean building height of 20 m, provides insights on advection processes. However, to assess the role of vertical mixing on spatial contrasts in air temperature, the response of ∆UHI to the vertical velocity variance at 238 m agl is shown in Fig 9b. This relation is even more clearly pronounced as can be seen from the smaller error statistics listed in Table 2. Given wind observations above the urban canopy layer (as here at Montsouris Park) are rarely available, the wind speed observations at the rural site Melun were also tested to have a full view of the ∆UHI

response to the regional regional winds. As the near-surface winds report higher uncertainty when predicting the ∆UHI intensity, it is concluded that turbulence observations inside the urban boundary layer show a closer link to the insensity the spatial constrar air temperature processess that drive the ∆UHI development. The curves in Fig 9b represents the best non-linear model fitted to the data collected during the nights with a LLJ event, using the air temperature data collected at QUALAIR-SU and by data collected by the IOP station located at Boulevard des Capucines (see Section 2.4). The similarity between the two curves indicates that the relationship between ∆UHI and $\sigma^2_w$ is preserved even when using QUALAIR-SU data collected at roof-level."

[Figure]

For Fig a) the dashed curve follows the equation UHI = 3((ws-1.2)$^{-0.5}$), while in Fig b) the equation is UHI = 4((ws-1.1)$^{-0.8}$). Wind speeds are stronger at the urban Montsouris Park (sensor at 25m above the ground) than at the rural site Melun (sensor height at 10m above the ground). For the current study, no σ2w has been conducted at Melun.

2. Generally, the study provides a lot of details. It would be helpful for the reader to provide some guidance on how the details relate to the broader context. For example, in the methods section, start each sub-section with an introductory/ summary sentence, so that the reader can pay attention to the details of interest. (e.g. start Sect. 2.2 with something like "Two DWLs, one in the city and one in a suburban are, were used to obtain vertical profiles of horizontal wind speed and vertical wind speed variance."

We agree that this modification will improve the clarity of the story for the reader. Particularly, the subsections 2.2 (p6, line 143) and 3.1 (p10, line 266) have been updated for clarity. The authors consider that the other subsections of the Methods section start with a clear sentence about the details related to the broader context.

3. The time periods used in this study are a bit confusing. It seems like the lidar analysis is based on the core period 15 June 2022 to 31 August 2022, but other time periods for validations are mentioned multiple times. Also, the PANAME initiative is not really introduced. Is the lidar study period part of this initiative or its Intensive Observation Period?

The time periods have been reviewed but no inconsistencies were found. The only different mentioned time period is on Page 5, line 124 (original manuscript): "Using a

comprehensive data set spanning from 2006 to 2022…". This is used to give the general context of the synoptic conditions of the study area. However, the text has been edited to ensure the clarity for the reader (p6, line 136) .

The Doppler Lidar (DL) installed at the urban site was deployed in January 2022 as part of the activities of the PhD project of Jonnathan Céspedes. The PANAME initiative started in the summer of 2022 (including multiple IOPs (Intensive Observation Period). In the framework of PANAME the DL at both the suburban and the urban sites are included as a key source of information given they provide continuous wind and turbulence (at QUALAIR) profile observations. The manuscript has been updated with a clear introduction to the PANAME initiative (p6, line 143) .

Specific comments

1.  The introduction is well-researched, but can be a little more concise and details not relevant to the study could be omitted.

    After reviewing the content of the introduction, the authors consider that on Page 2 between lines 36-47 details related to the previous locations around the world where the LLJ has been observed could be edited to be more concise. The introduction has been updated according to this comment.

2.  L. 160: Since the vertical velocity variance is highly sensitive to the averaging interval and sampling frequency, it might be worth pointing these out and giving an assessment about which parts of the turbulence spectrum are captured/ omitted with the used strategy.

    Thank you for pointing this out, certainly, the sampling frequency is decisive for estimating the vertical velocity variance. In the current study, the vertical stare scan records data continuously for 5-min in every 30-minute period. We consider that this is sufficient for the purpose of the investigation because an accurate sampling of the turbulence spectrum is beyond the scope of the study. The description of the scan strategy has been edited in the manuscript to highlight the 5-min of continuous vertical stare (P7, line 172).

    Based on experiments conducted in subarctic regions where the load of aerosols tends to be low, Yang et., (2019)[1] conclude that a 10-min vertical stare scan per hour is the required sampling interval to detect large-scale turbulence. While in urban areas where the content of aerosols is higher and unstable conditions are frequent, Bonin et al., (2018)[2] showed that representative turbulence can be detected at night with vertical stare sampling over 4 min per hour, allowing the detection of the mixing height layer based on the vertical velocity variance.

    On the other hand, by comparing theoretical and experimental approaches, Banakh et al., (2021)[3] showed that during nocturnal stable boundary layer conditions and the presence of a LLJ, 8 min of vertical stare measurements every hour are sufficient to identify the turbulent patterns produced below the jet core. They showed that vertical velocity oscillations are associated with the LLJ presence, and such oscillations can take between 30 min and 1 hour.
* * *
[1] https://doi.org/10.1002/met.1951
[2] https://doi.org/10.1175/JTECH-D-17-0159.1
[3] https://doi.org/10.3390/rs13112071

3. L. 210 "Note that the minimum below the core height may not be captured correctly by the observations because no information is available < 238m agl in the instrument's blind zone.": Are there near-surface wind measurements available at the urban site to close this gap? Especially, since QUALAIR-SU is referred to as a supersite.

Wind speed sampled within the urban canopy or on the top of tall buildings tends to be strongly affected by the roughness elements and bluff body effects. Hence, the more in-depth analysis of surface or roof-based wind measurements is beyond the scope of this study. A novel shallow DBS scan and retrieval implemented for 2023 finds shallow LLJ cases with a core height below 240 m in about 75% of the cases from the SE.

4. It seems like most of the LLJ statistics are based on the measurements from the urban lidar, and the suburban lidar serves only to show that the LLJs are a regional phenomenon. Maybe mention that at an early point in the paper, so that the reader does not expect a detailed analysis from the suburban lidar.

Section 2.2 has been edited according to this comment. The following sentence has been included:

p8, line: 181: " The data recorded with this instrument are used in this study only to highlight the regional scale of the LLJ observed. A detailed analysis of this data collection will be the subject of future studies."

5. L. 269 "Here we assume that the $\sigma_{2w}$ observations at the first range gate (238m agl) of the DWL at the urban site provides a representative proxy for vertical mixing in the nocturnal urban boundary layer.": Is the aim getting a wind variance estimate as close as possible to the surface? I am a bit worried that the variance at a fixed height agl depends on the jet core height.

Yes, we estimate the $\sigma^2_w$ values as close as possible to the surface to study the impacts of the mechanical turbulence below the jet core on the canopy layer UHI intensity. By considering $\sigma^2_w$ values at a fixed height we aim to assess the magnitude of vertical mixing exerted onto the surface layer.

It is correct that the vertical velocity variance at this fixed level may be affected by the LLJ core height and its core wind speed as these characteristics modulate the impact on the near-surface mixing. In future studies, we aim to compare the sigma w results from this first lidar gate to turbulence observations from urban flux tower sites. However, this requires a more in-depth source area analysis and determination of the blending height of the sonic anemometer observations and is hence beyond the scope of the current study.

Technical corrections

Please be more consistent with introducing and using abbreviations (IO, ABL, …)

Abbreviations have been edited for consistency.

L. 65: access -> excess?

This word has been changed.

L. 153: could you provide a reference for the hard target method?

The hard target is a north alignment method developed by the Doppler Lidar manufacturer Vaisala. At the moment there is no public document available that can be used as a reference.

---

## Author Comment (AC2)

**Review of egusphere-2024-520 "The Paris low-level jet during PANAME 2022 and its impact on the summertime urban heat island"**

**Submitted to Atmospheric Chemistry and Physics.**

This manuscript uses Doppler wind lidar measurements from two sites (one urban and one suburban) for the period 15 June – 31 August 2022 to analyse the summertime characteristics of the nocturnal low-level jet over Paris. The characteristics of the jets identified are similar to previous studies of low-level jets over Europe. Using surface wind and temperature observations, the study also investigates the relationship between the jet and the urban heat island. Vertical velocity variance over the city at 238m agl from the lidar is found to be a better predictor of the urban heat island than 10m winds at nearby rural site. There is not necessarily a relationship between jet strength and the urban heat island with some strong jet nights leading to strong urban heat islands. This is undoubtedly an interesting dataset and the continuous nature of the observations allows a study of the jet evolution which is not usually possible with less frequent radiosonde observations. However, I have some questions outlined below which need considering before the manuscript can be published.

The authors thank the referee for the detailed review of our paper and for the important feedback which is key for improving the quality of the manuscript. Modifications on the manuscript are highlightes with red text and here in this document we included the page and number line of the track changes manuscript.

**Major comments**

1) I did sometimes struggle to see what the novelty was in the results – the behaviour of the LLJ seems broadly in line with what would be expected based on other studies at similar latitudes / locations. The novelty seems to be around the interaction of the LLJ and urban environment, but the results seem to indicate the LLJ is not strongly affected by the urban land surface. The main result on the UHI seems to be the link with σw 2 which is irrespective of the LLJ. It would be good to focus on highlighting the novelty in a revised manuscript and particularly why the focus on the LLJ rather than UHI more generally.

The Conclusions section has been edited to highlight the original contribution of this work more clearly. Additionally, we list the key features below:

- For the first time, continuous Doppler Lidar measurements have allowed the acquisition of high-resolution vertical profiles of wind and turbulence in the Paris City center. This is an environment in which profile observations within the atmospheric boundary layer are urgently needed for the evaluation of high-resolution modeling (NWP, LES), and for gaining an improved understanding of the links between the synoptic background flow and the surface-driven processes for numerous applications, such as urban hydro-meteorological studies or air pollution transport.

- The adaptation and implementation of an algorithm for the automatic detection of the LLJ allow for the characterization of the phenomenon over an entire summer period. This is novel because no information had been available on the LLJ characteristics in

the Paris region thus far, with previous LLJ studies often limited to a few case studies. We find that the LLJ is a frequent nocturnal phenomenon (70% of nights) over Paris, and the results provide a detailed description of its speed and height in relation to flow dynamics (wind direction). While mechanical turbulence can also be produced by other mechanisms, our results reveal that the nocturnal LLJ is an important driver for nocturnal mixing processes in the UBL, with a clear impact on near-surface conditions.

- This study deploys an original approach by linking a regional-scale phenomenon (the LLJ) to the near-surface heat distributions (ΔUHI), through the Atmospheric Boundary Layer (ABL) dynamics. Usually, the urban canopy layer UHI is studied only by analyzing near-surface variables. Our study highlights that considering the 4D variability of ABL dynamics provides valuable insights into the drivers for mixing and advection processes that impact the heat distribution near the surface.

- As Section 3.7 discusses, cloudiness and surface wind speed both exert control of the UHI intensity. Selecting only cloud-free days from the study period, we isolated the surface wind speed control (Fig. 9a). However, another key contribution of this work is presented in Fig. 9b. We found that vertical mixing presents a better relationship than surface wind speed for the UHI intensity, allowing for a more accurate prediction of the UHI.

- While the vertical mixing in the atmosphere can be induced by other synoptic conditions different from a LLJ, Fig. 9b shows that moderate and strong vertical mixing is more likely to be present for nights with a LLJ event.

2) Much of the analysis focuses on the different classes of $\sigma w\,2$ however these are somewhat arbitrarily determined based on previous research. Can you justify the choice of thresholds? Does this depend on the location / the height at which the measurements are made? Is there a more objective way of determining what they should be?

As discussed in Section 3.2, in many descriptive studies about the LLJ, the authors present a profiles-oriented classification based on the $WS_{LLJ}$. In most cases, the wind speed intervals for such classification seem to be selected arbitrarily and vary between studies. Here, we propose a jet event-oriented classification based on the vertical mixing associated with each event. This system is proposed because our objective is to determine the relation between a LLJ event and the UHI which develops over a nocturnal period.

Given we are presenting a three-month data collection constituted by 30-minute average profiles, the data set is of sufficient size to apply a statistical approach and the vertical velocity variance thresholds are derived from the full dataset statistics. Given weak variance conditions were rather frequent in summer 2022 (50% of the nights with average $\sigma w\,2 < 0.1$), we find that these estimates do well represent the range of impacts of the LLJ on near-surface air temperature distributions. Interestingly, when applied to a different dataset (Paris summer 2023, not shown in this paper), the same thresholds still provide a meaningful characterisation in relation to the ΔUHI although the distribution of LLJ characteristics is different in terms of jet strength and altitudes. While the $\sigma w\,2$ classification proposed in this study based on the frequency distribution appears sufficiently meaningful, we are

investigating the links between variance indicators and other measures of atmospheric stability (temperature inversion intensity, bulk Richardson number) in future studies to see whether a more process-based classification can be determined.

**Minor comments**

1) p9, line 219. I do not understand what the difference in core time < 15 h means. Just above you stated that a LLJ event must have a LLJ detected in at least 4 consecutive profiles. Does this not mean the time between consecutive LLJ detections has to be 30 mins anyway? Please clarify.

A LLJ event is here defined to have a duration of at least 2h, defined by the time difference between start and end. However, in the 2h time window, we allow a maximum of 1 profile without LLJ detection. Hence the minimum jet event can consist of 3 profiles with LLJ detection over a period of 2h. In this case, there is one instance when the time difference is 1h between LLJ profiles. The sentence has been edited in the manuscript for clarity:

p9, line 231: "LLJ event detection: a LLJ event is considered as a coherent detection if lasts at least 2 hours (i.e. four 30-min averaged profiles). However, an event is valid if at least three profiles are detected over the course of this period with the following criteria for consecutive detections"

2) p10, line 270. You have chosen to use the first range gate (238m agl) to calculate $\sigma_w^2$. I have sometimes seen issues with data at the first range gate being a bit lower quality / noisier than further up. Did you see any issues with the data at the first range gate which might affect the results? Or to put it another way, are the results very dependent on whether you use the first gate, or say the 2nd / 3rd gate? In a sense this is a slightly arbitrary choice of height at which to calculate $\sigma_w^2$, so it would be good to know the results are not too sensitive to this choice.

We agree that the height at which we assess the vertical velocity variance is somewhat arbitrary as it is defined by the instrument characteristics and measurement setup. However, we have not noticed data quality issues on the first range gate. For comparison, we plot $\sigma_w^2$ derived at the first, second and third range gates (Figure below). These figures show that our conclusions are not especially sensitive to the choice of the range gate, as it illustrates that the height appears rather representative of the vertical mixing conditions in the nocturnal urban boundary layer. Of course, we acknowledge that representativity is especially challenged for LLJ at core altitudes < 250 m agl. We are working on novel scanning strategies and measurement setups to reduce this uncertainty in future studies. A comment was added on the Section 3.2..

[Figure]

[Figure]

3) p12, line 291. Here, and throughout the manuscript, you give very precise values for the heights of measurements (in this case 238m). In reality, given the range gates of the lidar are 75m, it is hard to justify this level of accuracy.

The length of the range gate refers to the size of the probing volume. The position of the measurement corresponds to the center of this volume, and can be accurately determined by the wind lidar with a sub-meter accuracy. Please note that the display resolution of the analysis is 25 m which is derived from the physical range gate resolution of 75 m. The core height statistics are discussed relative to the height above ground level. Then, the accuracy of the numbers makes sense considering that the height above ground level of the first range is 238 m. Also, some statistics like mean and median are provided as integers.

4) p13, Figure 4. The figure mentions classes 1, 2 and 3. These are not actually defined in the text. I assume these correspond to the low, medium and high vertical mixing classes?

Yes, those classes correspond to the low, medium, and high vertical mixing classes. The caption figure has been edited for clarification with the following sentence:

p13: "In the legend class 1 (red), 2 (yellow) and 3 (blue) correspond to low, medium and strong vertical mixing classes, respectively."

5) p13, line 323. I wonder why you don't also give a value of the wind speeds reported in the Karipot et al (2009) study? It's hard to compare meaningfully with the current results otherwise.

The wind speed values from Karipot et al (2009) have been included for clarity (see p14, line 343).

6) p14, line 344. "influence" -> "influences"

This has been modified in the manuscript.

7) p14, lines 348-350. I wonder if this sentence needs some qualification? I agree σw 2 is related to the surface layer turbulent heat fluxes in a convective boundary layer. I'm not sure this is necessarily the case in a stable boundary layer. It is also certainly true that shear generated turbulence can lead to vertical velocity variance and so surface layer turbulent heat fluxes are not solely responsible for σw 2 .

Thank you for pointing this out. In the cited studies, the decay and increase of $\sigma^2_w$ are linked to the diurnal cycle of the turbulent sensible heat flux that describes the surface-driven buoyancy in the CBL. In fact, Barlow et al 2015 and Banta et al. 2006 reveal with an analysis of the vertical velocity skewness that the LLJ is indeed the source of turbulence in such conditions, forming an "upside-down" boundary layer structure. For clarity, the sentence has been edited as follow:

p15, line 371: "In general, the shape of the three median profiles agrees with those reported in previous studies (Banta et al., 2006; Bonin et al., 2015). In cities, unstable or neutral stratification and higher $\sigma^2 w$ are mantained by the added urban heat (Theeuwes et al., 2019)."

8) p16, line 379. "the IO mechanism is highly relevant .. in the Paris region". This does seem to be the case for the example you look at, but since you have deliberately chosen an example without any other synoptic forcing this is perhaps not surprising?

This sentence requires quantification to justify the relevance of the IO mechanism in the Paris region. 28 out of 49 of the studied jet events display a clear IO signature in the hodograph, indicating the IO mechanism was highly relevant for the LLJ formation in summer 2022. The sentence has been edited for clarification (see p17, line 402).

9) p16, line 380. "T = 2π f" should be "T = 2 π / f"

This has been modified in the manuscript.

10) p16, line 381. "θ=48.8466". Needs units (degrees). It also seems to be an unjustifiably precise number, particularly when you then give the period T as ~16 hours.

Thanks for pointing this out. The period remains as T ~16h while the latitude has been provided with two digits "θ=48.84" .

11) p16, line 384-385. "wind speed amplitude". I find this a confusing phrase. You really mean the amplitude of the variations / oscillations in wind speed. Wind speed amplitude

could just be the wind speed itself. I would suggest rephrasing to make sure your meaning is clear.

The term "wind speed amplitude" has been replaced by "wind speed oscillation" throughout the entire manuscript.

12) p16, line 385. "with a  clockwise change"

This has been modified in the manuscript.

13) p16, line 389. "cases  do not show"

This has been modified in the manuscript.

14) p17, line 401. "decay  from about"

This has been modified in the manuscript.

15) p17, lines 402-403. This sentence is misleading. As written, it implies that the CBL during the morning is responsible for the gradual decay in jet core wind speed overnight. This cannot be true. What I think you mean is that the onset of the CBL in the morning leads to the jet core wind speed starting to increase again and hence is responsible for the minimum wind speed being right before sunrise.

The formation of the CBL enhances the vertical motions and increases the surface-driven buoyancy, breaking the atmospheric stratification and reducing the intensity of the wind speed at the level of the jet core. The sentence has been edited to express this more clearly:

p17, line 425: "The WSLLJ gradually starts to decay from about 4 h after sunset until it reaches a minimum right before sunrise, which is associated with the jet dissipation givenis the formation of the convective boundary layer during the morning."

16) p18, Figure 8b. I assume the number at the start of the line is a wind direction, but the caption doesn't actually say what. Is it the wind direction at sunset or at jet formation? I assume this is what WDLLJ SS means in the title? Again, not actually defined anywhere.

Thank you for pointing this out. The labels represent the wind direction values at sunset or the initial wind direction if the jet is formed in the middle of the night. The caption of the figure has been modified for clarification.

17) p19, lines 425-427. This sentence is confusing "These cases present differences .. regarding the 11 and 13 August…" Which cases? Differences compared to what? Are you trying to say that the 11 and 13 August are different to the other cases?

The sentence has been edited to ensure the clarification:

P 19, line 453: "A different situation is observed for the seven jets detected between the 06-13 August. Persistent synoptic conditions are prevail for the first five detections between

06-10 August, with similar nocturnal evolution of core characteristics and vertical mixing. However, the last two detections on the 11 and 13 August (outliers in Fig.7) differ clearly from this period as they have short duration, low core height, weaker wind speed, and low vertical mixing"

18) p20, line 460. This sentence is wrong. You have written that UHI is proportional to $\sigma_w 2$ and $\sigma_w 2$ is proportional to ZLLJ. In the first case, the relationship is definitely not linear according to figure 9b, and in the second case it also does not appear to be true (though it's harder to judge from looking at the colour of the dots in figure 9b. I agree there is a relationship between these variables, but they are not proportional.

Certainly, the intention is to express that the UHI intensity presents a clear non-linear relationship with $\sigma_w{}^2$ which in turn is dependent on the core height of the jet. Perhaps using the concept of proportionality is not the right approach to express this idea, that is why the manuscript has been edited to ensure clarity for the reader.

p21, line 463:"Table 2 presents a summary of the average values of ΔUHI and the LLJ characteristics, corresponding to each $\sigma_w{}^2$ class, revealing that in general low values of vertical mixing are associated with strong ΔUHI and calm winds and shallow core heights, whereas strong mixing relates to weak ΔUHI and strong wind speeds and greater core heights. "

19) p21, line 475. "see  Section 2.4)"

This has been modified in the manuscript.

20) p22, line 495. "such as  height, …"

This has been modified in the manuscript.

---

## Author Response (AR2)

**RESPONSE TO REFEREE NOMINATION & REPORT:  REFEREE NO. 1**

Thank you for responding to the comments and revising the manuscript.

The authors thank Anonymous Referee # 1 for their positive and valuable feedback that contributed to the improvement of the manuscript.

I have only one technical request: Is it possible that Fig. 9a was not updated? The authors response and the new caption suggest that measurements from Montsouris park are shown, but the Figure still shows the measurements from Melun.

In fact, the figure was not updated in the revised manuscript, it was a compilation mistake. The figure and its caption have been updated as follows:

[Figure]

Figure 9. Relations between the nocturnal average of ΔUHI intensity for all cloud-free nights in the study period and a) 10 m agl wind speed at Montsouris Park, an urban reference site. The dots are colored by wind direction at Montsouris (10 m agl). The dashed black curve is the best fit to the data: y = 4((x-1.1)$^{-0.8}$), and it follows the empirical relationship described by (Oke, 1973), and, b) the vertical velocity variance (σ$^2_w$) at 238 m agl. The dots are colored by the LLJ core height above the ground. The curves represent the best non-linear fit (y = a log(x) + b) found using data collected at QUALAIR-SU (*T rooftop*) (a = -1.30, b = 0.84) and at Boulevard de Capucines (*T street-level*) (a = -1.51, b = 0.77) observations, respectively (see Section 2.4). The background shading indicates the LLJ classes described in Section 3.2. In all subplots, dots and triangles represent nights with and without LLJ events, respectively.

**RESPONSE TO REFEREE NOMINATION & REPORT:  REFEREE NO. 2**

The authors have carefully addressed the comments from the two reviewers and as a result the clarity of the manuscript is much improved. I am happy to recommend acceptance subject to a few typographic / language errors which have been introduced with the changes.

The authors express their gratitude to Anonymous Referee #2 for their valuable feedback, which has improved the clarity and overall quality of this manuscript.

L141 - "the common of synoptic conditions" -> "the common synoptic conditions" or perhaps better would be something like "the typical synoptic conditions"

This has been modified in the manuscript following the suggestion of the Referee.

L147-149. "The PANAME initiave is an unprecedented converge

of multidisciplinary scietific investigations that promotes the synergy of numerous research projects that investigate the Paris urban environment in relation to weather, climate, air quality, and impacts on human health." is a bit of a confusing sentence, aside from having a couple of spelling mistakes. Something better might be "The PANAME initiative is an unprecedented programme bringing together a collection of multidisciplinary scientific projects that investigate the Paris urban environment in relation to weather, climate, air quality, and impacts on human health."

This has been modified in the manuscript following the suggestion of the Referee.

L176 - Replace both occurrences of "other" with "another".

This has been modified in the manuscript following the suggestion of the Referee.

L234 - "if lasts at least 2h" -> "if it lasts at least 2h"

This has been modified in the manuscript following the suggestion of the Referee.

L294 - "observations … is representative" -> "observations … are representative"

This has been modified in the manuscript following the suggestion of the Referee.

L456 - "conditions are prevail" -> "conditions prevail"

This has been modified in the manuscript following the suggestion of the Referee.

Figure 9 - please check the figures. The caption and text say the figures have been changed to show data from Montsouris, but the figure labels still say Melun.

In fact, the figure was not updated in the revised manuscript, it was a compilation mistake. The figure and its caption have been updated (see figure and caption above).

Figure 9 caption "Both curves represent the equation y = -1.39 log(x) + 0.84". The curves are different so how can they both represent the same equation? There are no free parameters in the equation.

See figure and caption above. In the fitting procedure used by the authors, no free parameters are present in the equation.

Table 2 caption "urban and rural surface wind speed at Melun" - should this be "urban wind speed at Montsouris nd rural wind speed at Melun"?

This has been modified in the manuscript following the suggestion of the Referee.

L545 "variability of LLJ" -> "variability of LLJs"

This has been modified in the manuscript following the suggestion of the Referee.